# Discovering and Leveraging Entropy-Complexity Relationships for Efficient Large Language Model Reasoning

## Abstract

Large Language Models (LLMs) suffer from "overthinking" — generating excessive reasoning chains even for simple problems, leading to computational inefficiency and potential accuracy degradation. To address this inefficiency, we systematically show that response entropy serves as an effective intrinsic measure of problem complexity in LLM reasoning. Our key insight is that token-level entropy during generation provides a principled signal for complexity assessment, where low entropy indicates high confidence suitable for direct answers, while high entropy signals the need for detailed reasoning. Building on this entropy-complexity relationship, we propose a novel two-stage training framework for adaptive reasoning. In Stage 1, we use Supervised Fine-Tuning (SFT) on No-Thinking exemplars (concise direct answers without explicit reasoning) to endow the model with concise answering capability. In Stage 2, we perform offline Proximal Policy Optimization (PPO) with an entropy-aware reward function to train models to dynamically select between concise and full reasoning modes based on problem complexity. This offline approach offers greater stability and efficiency compared to online RL methods. Experiments on MATH500, AIME24 and GPQA benchmarks demonstrate that our method significantly reduces response length while maintaining accuracy, validating entropy as both a diagnostic tool and training signal for efficient LLM reasoning. Our code is available via `https://anonymous.4open.science/r/Efficient-Reasoning-8BA6`.

## 1 Introduction

Large Language Models (LLMs) have achieved remarkable success in complex reasoning tasks, yet they exhibit a critical inefficiency: generating extensive intermediate reasoning chains even for problems that could be solved directly. This *overthinking phenomenon*—for instance, producing a 200-token reasoning sequence for "2+3=?"—represents a fundamental computational waste. It leads to higher latency, increased processing costs and potential accuracy degradation when excessive reasoning introduces errors. Even state-of-the-art models like OpenAI's o1 (OpenAI, 2024) and DeepSeek-R1 (DeepSeek-AI et al., 2025) suffer from this one-size-fits-all approach, generating verbose reasoning chains regardless of problem complexity.

To mitigate this, several approaches have been proposed. Inference-time methods like UnCert-CoT (Zhu et al., 2025) and CoT-Valve (Ma et al., 2025b) assess problem difficulty at runtime to dynamically allocate computational resources. However, this introduces undesirable overhead during inference. Other methods focus on training-time optimizations. AdaptThink (Zhang et al., 2025) utilizes online reinforcement learning, which is computationally intensive and can be unstable. More recently, O1-Pruner (Luo et al., 2025) introduced an offline training paradigm, but it relies on only length-based heuristics for complexity, lacking a fine-grained understanding of problem difficulty. These existing methods highlight a clear need for a more principled and efficient approach to adaptive reasoning.

In this paper, we show that response entropy is a reliable intrinsic signal of problem complexity and answer accuracy in LLM math reasoning. Through systematic empirical analysis, we find that token-distribution entropy during generation correlates strongly with both difficulty and accuracy.

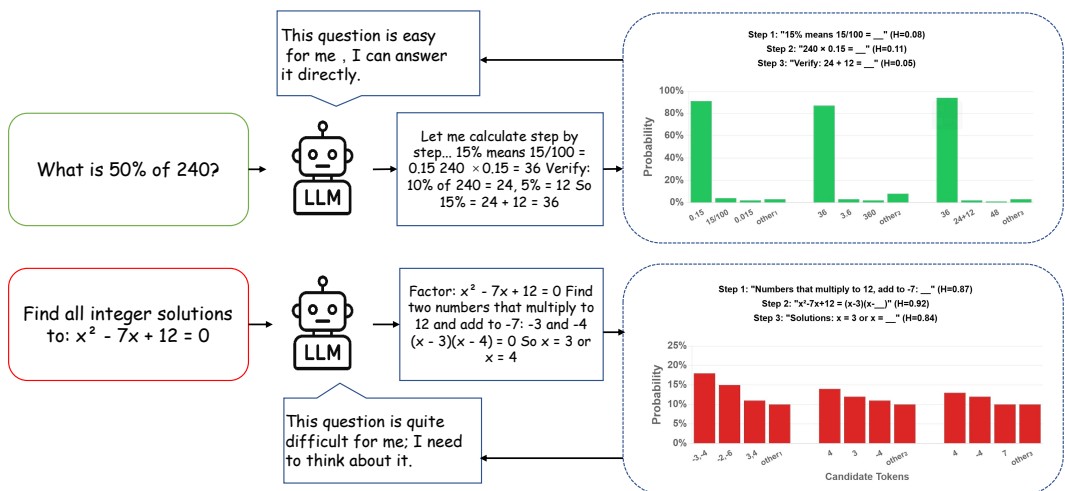

Figure 1: Entropy-driven adaptive reasoning: low-entropy for simple problems (Bottom), high-entropy for complex problems (Top).

Building on this insight, we introduce an entropy-driven framework that dynamically selects between abbreviated NoThinking and full chain-of-thought (Thinking): low-entropy predictions indicate high confidence and simple cases suited for direct answers, whereas high-entropy predictions flag complex cases that benefit from explicit reasoning. This provides a principled, continuous basis for adaptive reasoning without annotations or discrete thresholds.

Our approach employs a two-stage training paradigm combining the stability of supervised fine-tuning with the optimization power of offline reinforcement learning. In Stage 1, we train the model using supervised fine-tuning on NoThinking exemplars—samples where explicit reasoning chains within <think> tags are replaced with minimal placeholder tokens. This establishes the model's capacity for abbreviated reasoning without degrading performance. In Stage 2, we perform offline Proximal Policy Optimization (PPO) to reinforce the conditional selection of NoThinking versus full reasoning based on continuous entropy assessment, ensuring computational efficiency for simple problems while preserving comprehensive reasoning for complex scenarios.

Our empirical evaluation demonstrates the effectiveness of this approach across multiple reasoning benchmarks. On MATH500, AIME24 and GPQA datasets, our method achieves substantial efficiency gains while preserving accuracy on complex reasoning tasks. Specifically, we observe meaningful token reduction and improved computational efficiency, validating our core hypothesis that entropy-driven complexity assessment can effectively guide adaptive reasoning decisions.

This work makes three key contributions to efficient reasoning in LLMs:

- We systematically validate that response entropy serves as an effective intrinsic measure of problem complexity in LLM reasoning. Our experiment also reveals the cause of this phenomenon: the token distribution in the generated responses shifted toward higher entropy, providing a universal basis for assessing the difficulty of mathematical reasoning.

- We demonstrate that offline training combining SFT-generated NoThinking samples with PPO optimization achieves stable and efficient learning of reasoning mode selection.

- We conducted experiments on multiple models of varying sizes (1.5B, 7B, 8B) and architectures (Qwen, Llama), achieving the best results compared to other methods of the same type on the MATH500, AIME24 and GPQA test sets.

## 2 RELATED WORK

**The Overthinking Problem.** Recent reasoning models like OpenAI's o1 (OpenAI, 2024) and DeepSeek-R1 (DeepSeek-AI et al., 2025) achieve remarkable performance but generate verbose

reasoning chains regardless of problem complexity, which is known as *overthinking phenomenon* (Chen et al., 2025; Fan et al., 2025; Zhu & Li, 2025). Many recent studies (Zeng et al., 2025; Su et al., 2025) try to understand how generation length affects the accuracy of LLMs and reveal that deliberative reasoning capability by extremely long reasoning chains does not consistently improve model performance across diverse tasks, while Ma et al. (2025a) demonstrate that models can achieve competitive performance without explicit thinking for many tasks.

**CoT Compression and Adaptive Reasoning.** To mitigate the computational overhead of large language models in reasoning tasks, researchers have proposed various Chain-of-Thought (CoT) compression techniques (Wei et al., 2022; Yao et al., 2023; Besta et al., 2024). These methods aim to shorten the reasoning path by pruning, compressing, or distilling lengthy reasoning steps into a more compact form, thereby significantly reducing latency and computational costs while maintaining or even improving performance (Kang et al., 2025; Ma et al., 2025b; Lee et al., 2025; Luo et al., 2025). Further research has explored compressing discrete language-based reasoning steps into a continuous latent space, enabling more efficient thought processes by conducting reasoning within this denser representation (Cheng & Durme, 2024; Hao et al., 2024).

Another complementary line of research is adaptive reasoning, which centers on enabling models to dynamically allocate computational effort based on task complexity. For instance, a model can learn when to engage in deep, multi-step "slow thinking" and when to rely on more direct "fast thinking" to quickly arrive at an answer, thus striking a better balance between reasoning accuracy and efficiency (Han et al., 2025; Yi et al., 2025; Fatemi et al., 2025). This dynamic computation allocation allows the model to conserve resources on simpler problems while dedicating more capacity to solving more challenging ones.

# 3 PRELIMINARIES

## 3.1 ENTROPY

From an information-theoretic perspective, entropy measures uncertainty in a probability distribution. When LLMs generate tokens, high entropy indicates the model is uncertain among multiple plausible options. This typically occurs when: (1) the problem has inherent complexity with multiple valid reasoning paths, or (2) the model lacks confidence and requires additional reasoning to reduce uncertainty. We hypothesize that *response entropy serves as a proxy for problem complexity*. Unlike existing methods (human annotation, accuracy-based metrics, or answer length) that suffer from subjectivity, computational cost, or unreliability, entropy provides an intrinsic, continuous and computationally efficient measure.

## 3.2 EMPIRICAL VALIDATION

### 3.2.1 TOKEN-LEVEL ENTROPY EXTRACTION

During the generation process of an LLM, each token is sampled from a predicted probability distribution over the vocabulary. The entropy of this distribution captures the uncertainty associated with that token:

$$H_t = -\sum_i p_{t,i} \log p_{t,i} \tag{1}$$

where $p_{t,i}$ denotes the probability assigned to token $i$ at position $t$. High entropy indicates a flat distribution (i.e., uncertainty), while low entropy implies strong model confidence in its prediction. For each generated sequence, we calculate the entropy at every token position. Tokens with entropy greater than a threshold are selected as key tokens.

Considering that language model outputs may contain a significant number of low-entropy tokens with fixed formats (Wang et al., 2025), especially in mathematical fields, we propose a filtering method that retains only those tokens in the sequence whose entropy exceeds a certain threshold.

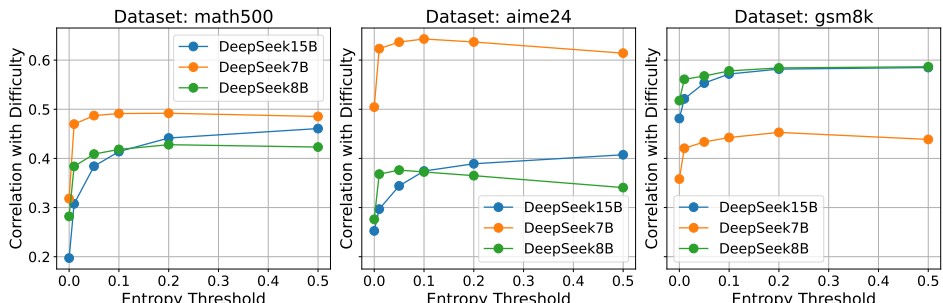

Figure 2: Entropy-difficulty correlation across models and datasets at different threshold.

We define the average entropy of the sequence as:

$$H_{\text{avg}} = \frac{1}{N} \sum_{i=1}^{N} H_t \cdot \mathbb{I}(H_t > \theta) \tag{2}$$

where $N$ is the number of tokens that exceed the threshold $\theta$ in the sequence, $H_t$ denotes the entropy of the $t$-th token, and $\mathbb{I}(H_t > \theta)$ is an indicator function that equals 1 when $H_t$ exceeds the threshold $\theta$, and 0 otherwise. This filtering strategy ensures that only high-entropy tokens are retained, thus mitigating the impact of low-entropy tokens on the model's performance. The average entropy of these tokens is then used to represent the model's overall uncertainty on that particular example.

### 3.2.2 FILTER-THRESHOLD ANALYSIS

Since datasets other than MATH lack manually annotated difficulty levels, to validate the rationality of the filter threshold selection as a difficulty proxy across more datasets, we propose a more general difficulty metric measurement method that does not require manual annotation, based on the optimized objective. We define

$$D = L_s - A_s \tag{3}$$

where $L_s$ and $A_s$ denote the normalized values of average accuracy and average length obtained by multiple sampling for each problem on the target dataset. The accuracy and length of the unprocessed, raw inference model were obtained through multiple sampling runs on the corresponding dataset. For verification of the formula's validity, see Appendix A.1.

We validated the entropy-complexity relationship across three models (DeepSeek-1.5B, 7B, 8B) and three datasets (MATH500, AIME24, GSM8K) at candidate thresholds $\tau \in \{0, 0.01, 0.1, 0.2, 0.5\}$.

Analyzing the experimental results in Figure 2, we observed that excessively low filtering thresholds resulted in poorer proxy difficulty correlation, while excessively high thresholds also yielded worse outcomes in certain scenarios. Based on the performance across the three datasets, we selected 0.2 as the filtering threshold.

The above experimental results demonstrate that, when selecting an appropriate filtering threshold relative to the dataset, the average entropy of model responses serves as an effective indicator reflecting reference difficulty. Based on Figure 2, we identify 0.2 as the filtering threshold on MATH500, where uncertainty correlates most strongly with difficulty. We then analyze its relationship with manually annotated difficulty (Figure 3) and adopt this threshold for subsequent fine-tuning on the MATH training set. We conducted further research into the causes of this phenomenon and found that the increase in average entropy is primarily due to a higher proportion of high-entropy tokens in generated responses. That is, the distribution of responses has shifted toward tokens with higher entropy values. Details are provided in Appendix A.2.

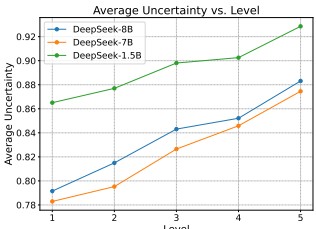

Figure 3: Average entropy in annotated difficulty levels.

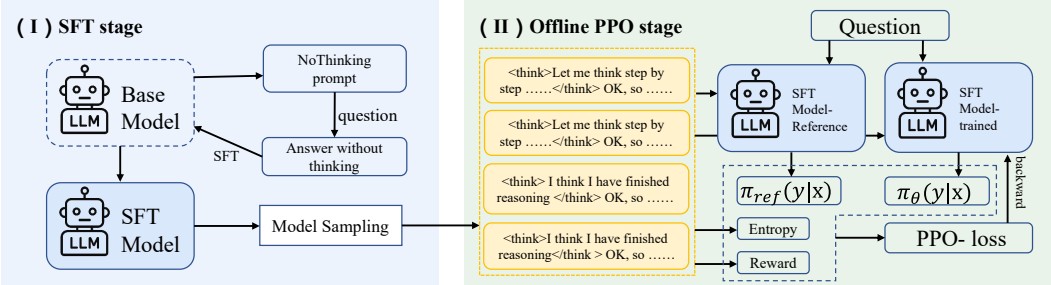

Figure 4: Our post-training framework involves two stages: The first stage employs an SFT approach, enabling the model to skip deliberation and provide answers directly with a certain probability. The second stage utilizes offline PPO to ensure the model's response style aligns with the complexity of the question.

## 3.3 Conclusion

Our experiments establish that token-level entropy serves as an effective intrinsic measure of problem complexity in LLM reasoning. This finding provides the theoretical foundation for our adaptive reasoning framework, enabling models to dynamically adjust their reasoning depth based on problem difficulty without external annotations.

## 4 Methodology

Our methodology is designed to equip LLMs with an adaptive reasoning capability, allowing them to dynamically balance computational efficiency and response accuracy. The core of our approach is a two-stage training process. We begin with Supervised Fine-Tuning (SFT) to teach the model a NoThinking shortcut for simpler queries. This is followed by an offline Proximal Policy Optimization (PPO) stage, which refines the model's ability to choose between immediate answers and deliberative reasoning based on the estimated difficulty of the problem, using response entropy as a key indicator. This offline approach significantly reduces the training overhead associated with traditional online reinforcement learning methods.

This two-stage design is crucial for instilling the desired behavior in a controlled manner. The initial SFT stage efficiently imparts the foundational ability to generate direct, NoThinking responses. Subsequently, the offline PPO stage builds upon this foundation, training the model to strategically apply this ability in an entropy-aware manner. The overall workflow of our proposed framework is illustrated in Figure 4. The first stage focuses on teaching the model *how* to produce immediate answers, while the second stage teaches it *when* to do so.

### 4.1 Stage 1: Supervised Fine-Tuning (SFT)

To efficiently incorporate NoThinking responses and prepare the model for offline PPO fine-tuning, we first perform a Supervised Fine-Tuning (SFT) stage. We construct the SFT dataset by selecting 1,912 problems from the two lowest difficulty tiers (Level 1 and 2) of the MATH (Hendrycks et al., 2021) training set and applying a prompt-injection template to elicit direct No-Thinking outputs, following Ma et al. (2025a). Specifically, we close the reasoning block using `<think>\nOkay, I think I have finished reasoning.\n</think>\n\n` to preserve contextual coherence. For each question, we collect multiple candidates and set the shortest verifiably correct response (under the model's tokenizer) as the SFT target. This endows the model with the ability to answer directly without disrupting its reasoning capability. However, responses at this stage may still depend on question format rather than intrinsic uncertainty. We address this in the subsequent reinforcement learning phase. Detailed generation and training configurations are provided in Appendix B.2 and B.5.1.

## 4.2 STAGE 2: OFFLINE PROXIMAL POLICY OPTIMIZATION (OFFLINE PPO)

In the second stage, we further refine the model's behavior to smartly choose between generating a direct answer (NoThinking) or a reasoned one (Thinking). Our approach builds on the findings from Section 3, where we demonstrate a strong correlation between response entropy and problem difficulty. This provides a theoretical basis for using an entropy-aware reward function in our PPO training strategy. We modify the widely used accuracy-length reward by incorporating average entropy as part of the accuracy reward, while integrating characteristics of both response modes in the model. For direct responses, we aim for the model's answers to be deterministic and negatively correlated with uncertainty. Conversely, for deliberative responses, we seek rewards that are positively correlated with uncertainty. To achieve this, we adopt an off-policy PPO approach that employs a reward function and a baseline-corrected signal for stability. Detailed training configuration can be found in Appendix B.5.2.

**Offline Data Collection.** We view LLMs after Stage 1 as a reference policy model, denoted $\pi_{\text{ref}}$ and begin by generating a static dataset of diverse responses. For each problem $x$ in training set, we generate K candidate responses $\{y_1, \ldots, y_K\}$ by sampling from $\pi_{\text{ref}}$. This collection of responses, $\mathcal{D}_{\text{off}} = \{(x, y_i(x)) \mid x \in \mathcal{D}, i = 1, \ldots, K\}$, is used for the policy optimization phase.

**Reward Formulation with Advantage Baseline.** Our reward function is designed to provide a detailed feedback signal to the model. For a given question $x$ and response $y$, we first define a multi-component base reward, $R_{\text{base}}(x, y)$, as the sum of three distinct terms:

$$R_{\text{base}}(x, y) = w_1 R_{\text{acc}}(y) + w_2 R_{\text{eff}}(x, y) + w_3 R_{\text{diff}}(x, y) \tag{4}$$

The components are defined as follows:

**Accuracy Reward ($R_{\text{acc}}$):** Reward for correctness, where $\mathbb{I}(\cdot)$ is the indicator function.

$$R_{\text{acc}}(y) = \mathbb{I}(\text{is\_correct}(y)) \tag{5}$$

**Efficiency Penalty ($R_{\text{eff}}$):** Penalty for exceeding the average response length of $\pi_{\text{ref}}$, $\bar{L}_{\text{ref}}(x)$, which is pre-computed from the K sampled responses for question $x$.

$$R_{\text{eff}}(x, y) = -\max\left(0, \frac{\text{len}(y)}{\bar{L}_{\text{ref}}(x)} - 1\right) \tag{6}$$

Since direct responses are significantly shorter than those given after deliberation, this metric not only encourages the model to provide immediate answers but also incentivizes it to choose shorter responses when deliberation is possible.

**Difficulty-Adaptive Reward ($R_{\text{diff}}$)** A reward that encourages a response style appropriate to the question's difficulty, which is estimated by the model's average entropy $H_{\text{avg}}(x)$ (calculated as in Equation 2). Our analysis in Section 3 shows that lower entropy correlates with simpler questions. We therefore reward *NoThinking* for low-entropy questions and *Thinking* for high-entropy ones to encourage efficient problem-solving. The reward is defined as:

$$R_{\text{diff}}(x, y) = \mathbb{I}(\text{is\_correct}(y)) \cdot \begin{cases} (1 - H_{\text{avg}}(x)) & \text{if } y \text{ is NoThinking} \\ H_{\text{avg}}(x) & \text{if } y \text{ is Thinking} \end{cases} \tag{7}$$

This term rewards direct answers for simple (low entropy) questions and reasoned answers for complex (high entropy) ones.

To stabilize training, we compute a baseline reward, $\bar{R}_{\text{ref}}(x)$, by averaging the base reward over the 16 reference responses for each question.

$$\bar{R}_{\text{ref}}(x) = \frac{1}{K} \sum_{i=1}^{K} R_{\text{base}}(x, y_i'), \quad \text{where } y_i' \sim \pi_{\text{ref}}(\cdot|x) \tag{8}$$

The final reward signal used for optimization is the advantage of the response $y$ over this baseline:

$$R_{\text{adv}}(x, y) = R_{\text{base}}(x, y) - \bar{R}_{\text{ref}}(x) \tag{9}$$

**Off-Policy Objective.** We employ a PPO-style clipped objective function to optimize the policy $\pi_\theta$ using the offline dataset and our advantage-based reward. The objective function is formulated as:

$$L^{\text{OFF-PPO}}(\theta) = -\mathbb{E}_{(x,y) \sim \mathcal{D}_{\text{off}}} \left[ \min\left(r(\theta) R_{\text{adv}}(x, y), \text{clip}(r(\theta), 1 - \epsilon, 1 + \epsilon) R_{\text{adv}}(x, y)\right) \right], \tag{10}$$

where $r(\theta) = \dfrac{\pi_\theta(y|x)}{\pi_{\text{ref}}(y|x)}$ is the probability ratio between the current policy $\pi_\theta$ and the fixed reference policy $\pi_{\text{ref}}$. This allows us to prepare the required data at the beginning of training, thereby greatly reducing the training overhead.

## 5 EXPERIMENTS

### 5.1 EXPERIMENTAL SETUP

**Large Reasoning Models.** We evaluate three open-source reasoning LLMs from the DeepSeek-AI et al. (2025) model families with different sizes and base models: *DeepSeek-R1-Distill-Qwen-1.5B*, *DeepSeek-R1-Distill-Qwen-7B* and *DeepSeek-R1-Distill-Llama-8B*. These models encompass two mainstream large-model architectures: Qwen and Llama. Complete generation hyperparameters and prompting templates are provided in Appendix B.

**Dataset.** We evaluate our method using three widely adopted benchmarks: MATH500 (Lightman et al., 2023), AIME24 and GPQA (Rein et al., 2024). MATH500 is a subset of 500 problems sampled from the MATH dataset, covering algebra, calculus, geometry and number theory. This dataset is designed to test mathematical reasoning across a range of difficulty levels. AIME24 dataset contains problems from the American Invitational Mathematics Examination (AIME) 2024 and is a prestigious high school mathematics competition known for its challenging mathematical problems. GPQA comprises graduate-level STEM reasoning tasks. The GPQA Diamond dataset is a benchmark dataset focused on multi-domain knowledge question-answering, constructed by a professional research team.

**Baselines.** To validate the superiority of our method, we have selected the five representatively comparative methods. (1) **NoThinking**: The NoThinking method is a prompt injection technique introduced by Ma et al. (2025a) to enforce reasoning models to skip explicit thinking. (2) **Fast-Solving Prompt**: The Fast-Solving Prompt is a prompting technique wherein we instruct the model within the prompt to solve the given problem as swiftly as possible, hoping to achieve the desired reduction in reasoning length. (3) **SFT**: For the SFT method, we curated the training dataset by selecting the shortest correct solutions for each problem, also known as self distillation, ensuring that the model is exposed to examples that embody both accuracy and conciseness. These solutions were then used to train the model following the standard SFT pipeline. (4) **DPO** (Rafailov et al., 2023): For the implementation of DPO, we meticulously selected two of the shortest correct solutions to serve as the chosen samples, which exemplify efficiency and precision in problem-solving. Conversely, to represent the rejected sample, we opted for the longest solution available. (5) **O1-Pruner** (Luo et al., 2025): first estimates the reference model's performance through pre-sampling and then uses off-policy PPO-style fine-tuning to encourage the model to generate shorter reasoning processes under accuracy constraints.

**Evaluation Metric.** We employ the following average accuracy and average length to evaluate the performance of our methods. Accuracy is reported as Pass@1 using the Open-R1 framework (Hugging Face, 2025). Length is defined as the number of generated output tokens, excluding prompt tokens; tokens within `<think>` and `</think>` sections are included.

**Implementation Details.** Experiments are conducted on a cluster of 8 NVIDIA A100 GPUs. Detailed training configurations can be found in Appendix B.5.

### 5.2 EXPERIMENTAL RESULTS

#### 5.2.1 MAIN RESULTS

As presented in Table 1, our two-stage SFT+PPO framework consistently outperforms all baseline methods across the three models and datasets. Our approach achieves an optimal balance between accuracy and response length. For instance, with the DeepSeek-1.5B model, our method not only improves the average accuracy by 3.0% but also significantly reduces the average response length by 47.5%. In contrast, while the DPO method also enhances accuracy, its effectiveness in shortening response length is considerably less pronounced (a 9.2% reduction compared to our 47.5%). Other baselines, such as Original$_{\text{NoThinking}}$ and Fast-Solving Prompt, achieve substantial length re-

Table 1: Main experimental results across models and datasets.

| Method | MATH 500 | | AIME 2024 | | GPQA | | Average | |
|---|---|---|---|---|---|---|---|---|
| | Acc | Length | Acc | Length | Acc | Length | $\Delta$Acc | $\Delta$Length |
| *DeepSeek-R1-Distill-Qwen-1.5B* | | | | | | | | |
| Original$_{Thinking}$ | 85.2 | 5461 | 23.3 | 17231 | 39.4 | 9405 | - | - |
| Original$_{NoThinking}$ | 70.4 | 944 | 10.0 | 2635 | 13.6 | 782 | -18.0 | -86.4% |
| Fast-Solving Prompt | 78.2 | 3344 | 26.7 | 14424 | 36.9 | 9312 | -2.0 | -18.7% |
| SFT | 80.6 | 4965 | 23.3 | 17575 | 32.8 | 10322 | -3.7 | +0.9% |
| DPO | **84.8** | 4816 | **33.3** | 14219 | 34.8 | 9556 | +1.7 | -9.2% |
| O1-Pruner | 82.8 | 4635 | 30.0 | 16841 | 37.9 | 8869 | +0.9 | -7.7% |
| Ours | 84.2 | **3257** | **33.3** | **7250** | 39.4 | **5260** | **+3.0** | **-47.5%** |
| *DeepSeek-R1-Distill-Qwen-7B* | | | | | | | | |
| Original$_{Thinking}$ | 93.2 | 3956 | 56.7 | 12102 | 49.5 | 8292 | - | - |
| Original$_{NoThinking}$ | 79.6 | 723 | 23.3 | 2921 | 12.1 | 714 | -28.1 | -83.0% |
| Fast-Solving Prompt | 88.8 | **2738** | 50.0 | 12910 | **53.0** | 7992 | -2.5 | -9.2% |
| SFT | 89.8 | 3912 | 43.3 | 9799 | 48.0 | 10437 | -6.1 | +1.9% |
| DPO | 94.0 | 3227 | 56.7 | **9514** | 52.0 | 9056 | +1.1 | -10.2% |
| O1-Pruner | **94.8** | 3895 | **63.3** | 11401 | 49.5 | 8731 | **+2.7** | -0.7% |
| Ours | 92.4 | 3105 | **63.3** | 10230 | 49.0 | **6626** | +1.8 | **-19.0%** |
| *DeepSeek-R1-Distill-Llama-8B* | | | | | | | | |
| Original$_{Thinking}$ | 85.0 | 4390 | 46.7 | 13717 | 50.0 | 8977 | - | - |
| Original$_{NoThinking}$ | 65.8 | 964 | 13.3 | 5961 | 16.7 | 1283 | -28.6 | -73.4% |
| Fast-Solving Prompt | 78.2 | 3194 | 40.0 | 12029 | 48.0 | 9069 | -5.2 | -12.8% |
| SFT | 75.2 | **3170** | 40.0 | 13493 | 47.0 | 8815 | -6.5 | -10.4% |
| DPO | **88.4** | 4030 | 50.0 | 12874 | **51.5** | 8515 | +2.7 | -6.5% |
| O1-Pruner | 86.6 | 3821 | 53.3 | 11344 | 48.5 | 8537 | +2.2 | -11.7% |
| Ours | 85.8 | 3564 | **56.7** | **10126** | 49.0 | **7135** | **+3.3** | **-21.8%** |

ductions but at the cost of a severe drop in accuracy. On the DeepSeek-8B model, for example, Original$_{NoThinking}$ leads to a steep 28.6% decline in average accuracy. The SFT method exhibits unstable performance and, in some cases, even degrades performance. These results robustly demonstrate that our SFT+PPO framework effectively enhances the model's reasoning capabilities while significantly improving its efficiency, striking an excellent trade-off between accuracy and conciseness. Complementary experiments can be found in Appendix C.

## 5.3 MECHANISTIC ANALYSIS: EFFICIENT REASONING UNDER CONSTRAINTS

To understand how our approach balances computational efficiency and accuracy, we analyze the model's behavior across different problem complexities. Our findings reveal that under the constraints of offline PPO training, the model develops an effective strategy for resource management, achieving a favorable balance between token reduction and accuracy. All figures are experimented on DeepSeek-8B. We roll out for 8 times at each sample on MATH500 dataset.

### 5.3.1 NOTHINKING PROPORTION AND ACCURACY

Figure 5a indicates that after only SFT (Stage 1), the model already possesses the ability to skip thinking processes and output directly on a large scale. It can also automatically select output modes to some extent based on question difficulty. However, the simple SFT process cannot guarantee that the model's NoThinking outputs precisely match the actual difficulty of the questions. This leads to the model generating verbose outputs for some easy questions while excessively skipping the thinking process for difficult ones. After offline PPO fine-tuning (Stage 2), we observe that while the model's overall NoThinking responses decrease, both Thinking and NoThinking responses show improved accuracy (Figure 5b). Concurrently, the average response length reduces. This indicates

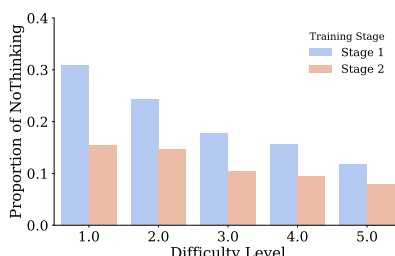

(a) Impact of reinforcement fine-tuning on the proportion of NoThinking responses.

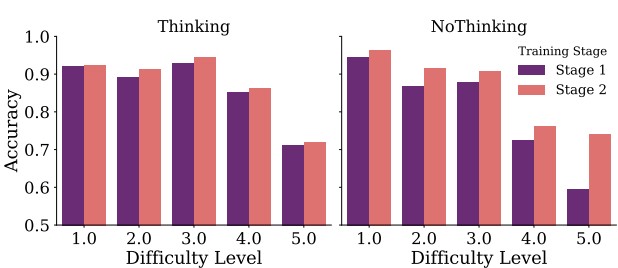

(b) Impact of reinforcement fine-tuning on accuracy.

Figure 5: NoThinking proportion and accuracy comparison on DeepSeek-8B model.

that NoThinking responses generated by the reinforcement-fine-tuned model better align with the distribution of question difficulty.

### 5.3.2 RESPONSE LENGTH DISTRIBUTION

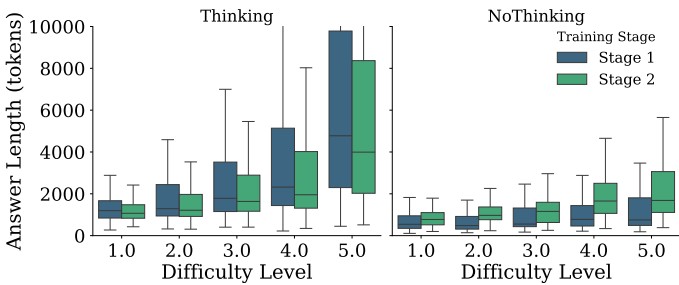

Figure 6: Response length distribution comparison on DeepSeek-8B model.

The primary driver of efficiency is a learned conciseness in Thinking responses. Figure 6 shows a substantial reduction in response length for Thinking outputs, especially for more complex problems (Level 4 and 5). This indicates the model learns to generate more efficient reasoning paths rather than simply avoid reasoning altogether. Meanwhile, the model appropriately increases the response accuracy for NoThinking queries by attempting to extend response length—particularly when dealing with more challenging questions.

## 6 CONCLUSION

This study successfully validates that token-level entropy can serve as an effective indicator of both problem difficulty and model confidence. Based on this finding, we develop and validate a two-stage training framework designed to mitigate the "overthinking" problem in LLMs, thereby enhancing their reasoning efficiency. In summary, we provide a practical and theoretically supported solution for building more efficient and intelligent language models. By learning adaptive reasoning strategies during the training phase, our method avoids inference-time overhead, opening new possibilities for deploying high-performance language models in resource-constrained environments.

## 7 LIMITATIONS

The foundational theory of difficulty-entropy applies only to mathematical reasoning tasks (MATH500, AIME24, GSM8K), and its universality across other domains where entropy patterns may differ significantly remains unclear. Our use of discrete entropy thresholds instead of continuous adaptation mechanisms may miss opportunities for finer-grained allocation of reasoning resources, and these thresholds require domain-specific tuning.

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

## USE OF LLMS

We express our gratitude to the large language models that assisted in the polishing of this paper. Their contributions have been instrumental in refining the text, ensuring clarity and precision in the presentation of our research.

## A  ENTROPY THEORY AND ANALYSIS

### A.1  THEORETICAL JUSTIFICATION: $D = L_s - A_s$ AS A DIFFICULTY PROXY

Consider a problem set $\mathcal{Q}$ with inherent difficulty function $d : \mathcal{Q} \to \mathbb{R}^+$. For each problem $q \in \mathcal{Q}$, let $a(q) \in [0, 1]$ denote the model's accuracy and $l(q) \in \mathbb{R}^+$ denote the response length. Under our adaptive reasoning framework, the model aims to minimize the following objective:

$$\mathcal{L}(\theta) = \mathbb{E}_{q \sim \mathcal{Q}} [l(q) - \lambda \cdot a(q)] \tag{11}$$

where $\lambda > 0$ controls the trade-off between brevity and accuracy.

**Assumption 1 (Difficulty-Length Relationship):** For an optimal model under objective $\mathcal{L}$, the expected response length increases monotonically with problem difficulty:

$$\mathbb{E}[l(q)|d(q) = d_1] < \mathbb{E}[l(q)|d(q) = d_2] \quad \text{for } d_1 < d_2 \tag{12}$$

**Assumption 2 (Difficulty-Accuracy Relationship):** The expected accuracy decreases monotonically with problem difficulty:

$$\mathbb{E}[a(q)|d(q) = d_1] > \mathbb{E}[a(q)|d(q) = d_2] \quad \text{for } d_1 < d_2 \tag{13}$$

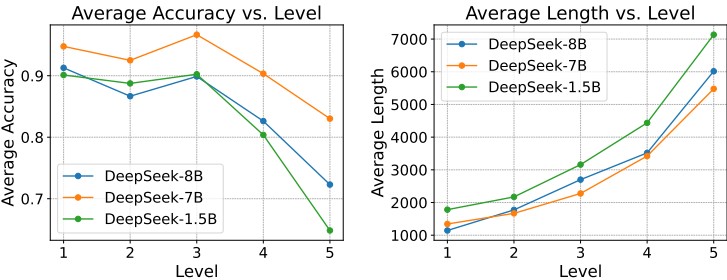

Figure 7: Accuracy Improvement After Entropy-Aware Fine-Tuning

Figure 7 presents the performance of DeepSeek models on the MATH500 test set, categorized by manually annotated difficulty levels from 1 (easiest) to 5 (hardest). Observing the two subgraphs reveals that across the MATH500 test set, all three large models exhibit a general trend of decreasing accuracy with increasing difficulty level, while answer length increases with difficulty. Through this experiment, we have briefly verified the two hypotheses proposed above.

**Theorem 1.** Under Assumptions 1-2, the standardized difference $D(q) = L_s(q) - A_s(q)$ is a monotonically increasing function of the true difficulty $d(q)$.

**Proof:** Let $L_s(q) = \frac{l(q)-\mu_l}{\sigma_l}$ and $A_s(q) = \frac{a(q)-\mu_a}{\sigma_a}$ be the standardized length and accuracy respectively, where $\mu$ and $\sigma$ denote means and standard deviations.

For two problems $q_1, q_2$ with difficulties $d(q_1) < d(q_2)$:

$$D(q_2) - D(q_1) = [L_s(q_2) - A_s(q_2)] - [L_s(q_1) - A_s(q_1)] \tag{14}$$

$$= [L_s(q_2) - L_s(q_1)] - [A_s(q_2) - A_s(q_1)] \tag{15}$$

$$= \frac{1}{\sigma_l}[l(q_2) - l(q_1)] - \frac{1}{\sigma_a}[a(q_2) - a(q_1)] \tag{16}$$

By Assumption 1: $l(q_2) > l(q_1) \Rightarrow l(q_2) - l(q_1) > 0$

By Assumption 2: $a(q_2) < a(q_1) \Rightarrow a(q_2) - a(q_1) < 0$

Therefore:

$$D(q_2) - D(q_1) = \underbrace{\frac{1}{\sigma_l}[l(q_2) - l(q_1)]}_{>0} - \underbrace{\frac{1}{\sigma_a}[a(q_2) - a(q_1)]}_{<0} > 0 \qquad (17)$$

Thus, $D(q)$ increases monotonically with true difficulty $d(q)$.

**Corollary 1.** Under the optimization objective $\mathcal{L}(\theta)$, problems with $D(q) < 0$ (high accuracy, low length) are empirically easier and benefit from NoThinking responses, while problems with $D(q) > 0$ require detailed reasoning.

This theoretical framework justifies using $D = L_s - A_s$ as a proxy for problem difficulty in our entropy-based adaptive reasoning approach, as it naturally emerges from the model's optimization dynamics rather than being an arbitrary heuristic.

## A.2 MATH500 ENTROPY ANALYSIS

To validate our hypothesis that token-level entropy serves as an effective indicator of problem complexity, we conducted a comprehensive analysis of entropy patterns across different difficulty levels. We sampled the MATH500 dataset 8 times and calculated the average, filtered out tokens with entropy below 0.1 to focus on meaningful reasoning content, excluding formatting elements and deterministic expressions that do not reflect actual reasoning uncertainty.

### A.2.1 ENTROPY DISTRIBUTION PATTERNS

Figure 8 illustrates the entropy distribution across five difficulty levels. The analysis reveals a systematic shift in the distribution as problem complexity increases. For difficulty level 1.0, the probability is concentrated in the lower entropy ranges (0.1-0.5), with 26.8% of tokens falling in this region. As difficulty increases to 5.0, this proportion decreases to 24.6%, while the high entropy region ($>1.0$) expands from 25.2% to 27.7%.

### A.2.2 MECHANISMS BEHIND ENTROPY INCREASE

The increase in average entropy can be attributed to three primary mechanisms:

**Increased Decision Complexity**    As shown in Figure 9, the entropy distribution heatmap reveals a clear rightward shift in the probability mass. Simple problems (difficulty 1.0) exhibit deterministic reasoning patterns where the model has high confidence in its next-token predictions. Complex problems (difficulty 5.0) require the model to consider multiple viable reasoning paths, resulting in higher prediction uncertainty.

**Reasoning Mode Transition**    The segmentation analysis in Figure 10 demonstrates a fundamental shift in reasoning modes. Low-difficulty problems predominantly employ pattern matching and direct application of known formulas (low-mid entropy: 0.1-0.5). High-difficulty problems require exploratory reasoning and hypothesis testing (high entropy: $>1.0$). The stacked area chart clearly shows this transition, with the high-entropy segment expanding 12.7-fold in absolute terms while its proportion nearly doubles.

**Statistical Evidence**    Figure 11 provides quantitative evidence for the entropy increase mechanism. The mean entropy shows a consistent upward trend with a 6.3% increase from difficulty 1.0 to 5.0. Notably, the coefficient of variation decreases from 0.629 to 0.619, indicating that while overall uncertainty increases, the distribution becomes more stable and predictable. This suggests that the model develops consistent patterns for handling complexity rather than random fluctuations.

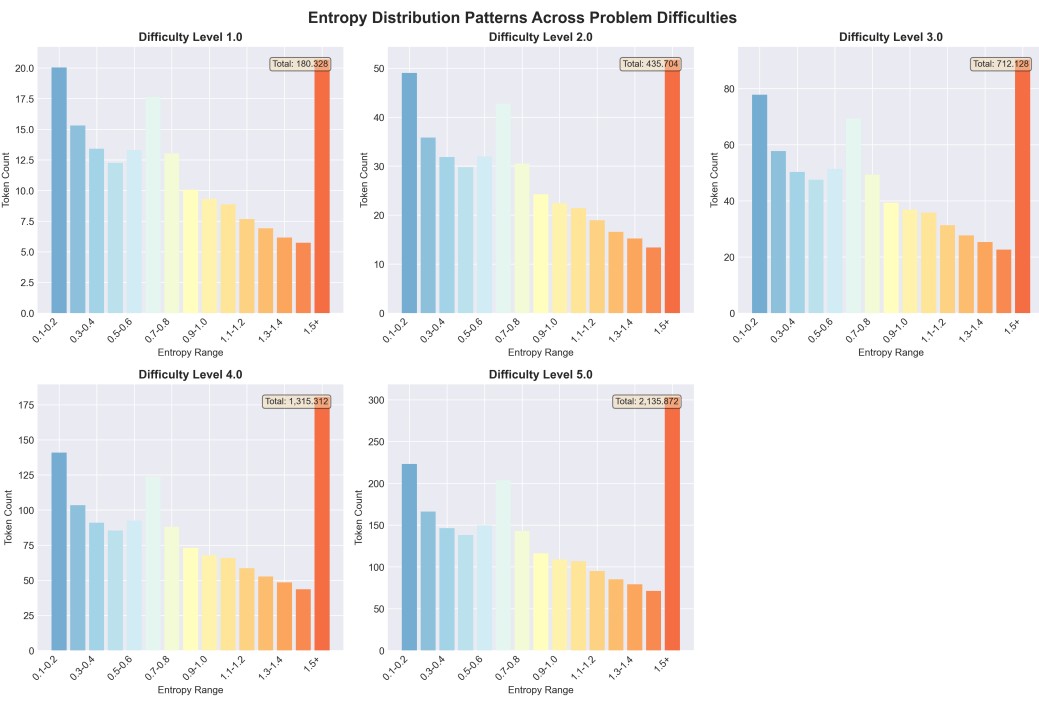

Figure 8: Entropy distribution patterns across problem difficulties. Each subplot shows the token count distribution across entropy ranges for a specific difficulty level.

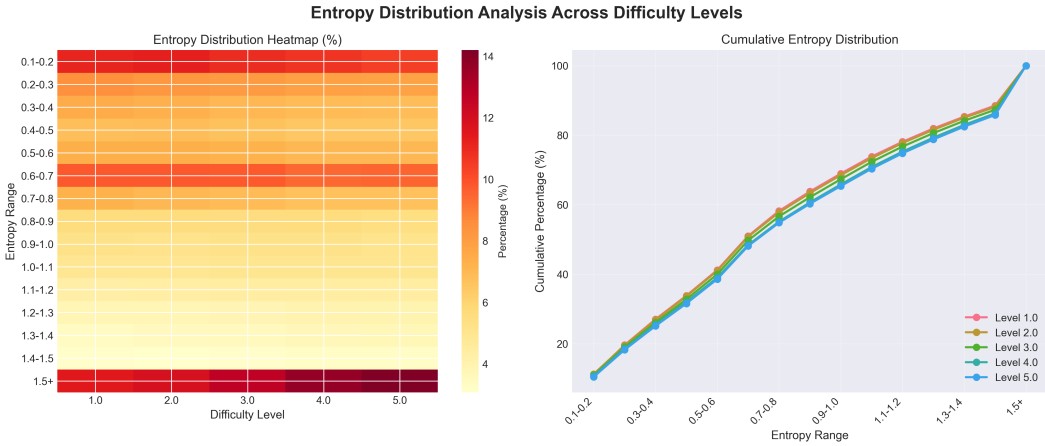

Figure 9: Entropy distribution heatmap and cumulative trends. **Left:** Heatmap showing the percentage distribution of tokens across entropy ranges for each difficulty level. **Right:** Cumulative distribution curves demonstrating the progressive shift toward higher entropy ranges.

### A.2.3 HIGH-FREQUENCY TOKEN ANALYSIS

An intriguing finding emerges from the analysis of high-frequency tokens (Figure 12). Core reasoning tokens such as "the", "I" and "that" maintain remarkably stable entropy values (1.77-1.90) across all difficulty levels, while their number of occurrences increases exponentially 35-fold from difficulty 1.0 to 5.0. This stability-frequency paradox indicates that:

1. The model's fundamental reasoning vocabulary remains consistent regardless of problem complexity

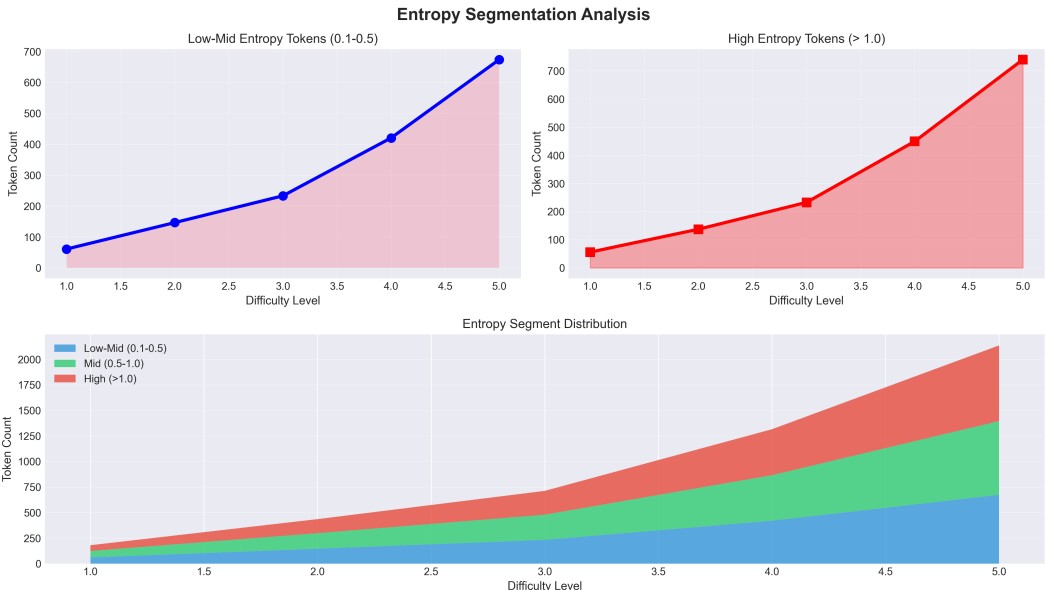

Figure 10: Entropy segmentation analysis. **Top left:** Growth of low-mid entropy tokens. **Top right:** Growth of high entropy tokens. **Bottom:** Stacked area chart showing the evolution of entropy segments across difficulty levels.

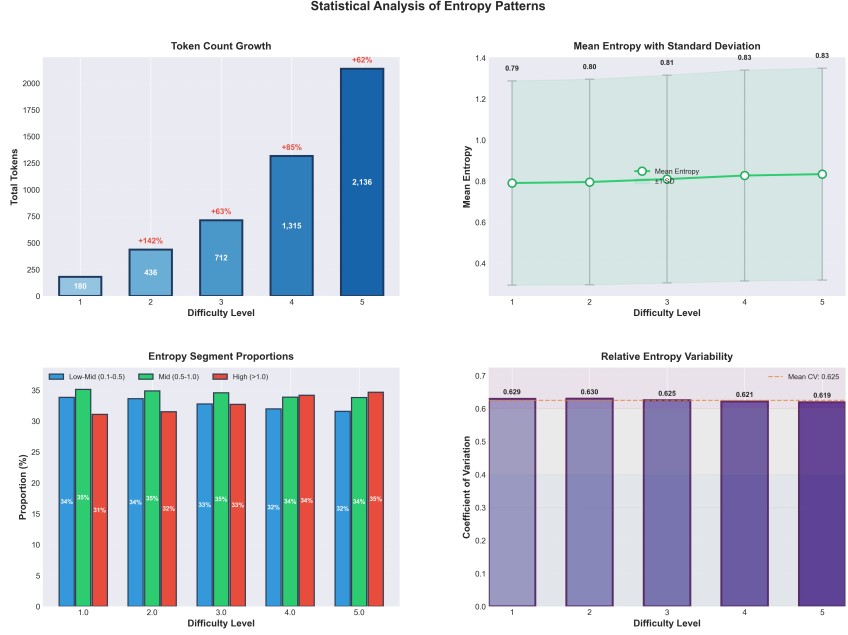

Figure 11: Statistical analysis of entropy patterns. **Top left:** Token count growth showing 12.7x increase. **Top right:** Mean entropy with standard deviation bands. **Bottom left:** Proportions of entropy segments. **Bottom right:** Coefficient of variation showing decreasing relative variability.

2. Increased difficulty manifests not through changes in individual token uncertainty but through the proliferation of decision points

3. The entropy increase is driven by the aggregate effect of more frequent high-entropy decisions rather than higher per-token uncertainty

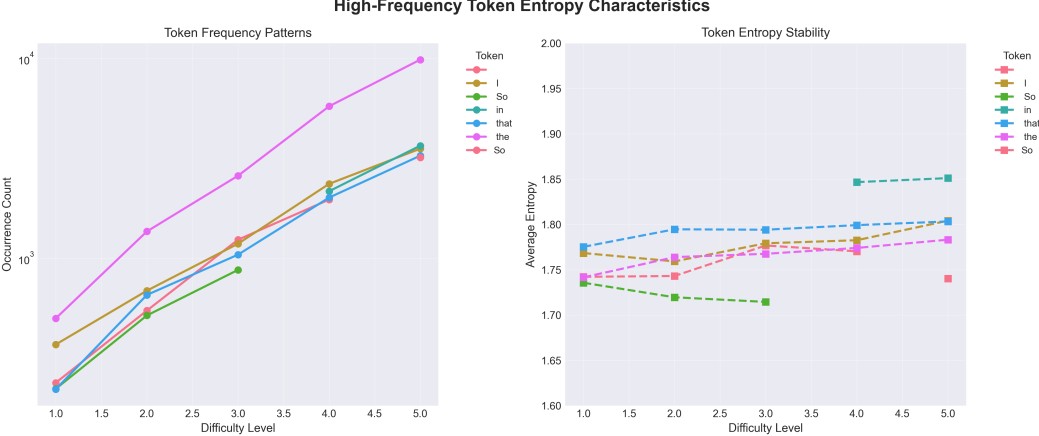

Figure 12: High-frequency token analysis. **Left:** Log-scale frequency patterns showing exponential growth. **Right:** Entropy stability of top tokens across difficulty levels, maintaining values between 1.7 and 1.9.

### A.2.4 IMPLICATIONS FOR ADAPTIVE REASONING

These findings provide strong empirical support for using entropy as a complexity signal in adaptive reasoning systems. The systematic relationship between entropy and difficulty, combined with the clear segmentation of entropy ranges, enables reliable identification of problem complexity during inference. The filtering threshold of 0.1 effectively separates meaningful reasoning tokens from formatting elements, while the natural boundaries at 0.5 and 1.0 provide clear decision thresholds for adaptive reasoning strategies.

The 6.3% increase in mean entropy represents a fundamental shift in the model's reasoning approach: from confident, deterministic processing of simple problems to exploratory, multi-path reasoning for complex problems. This validates our core hypothesis that entropy serves as an intrinsic measure of reasoning complexity, providing a principled foundation for our adaptive reasoning framework.

## A.3 CROSS-DATASET ENTROPY ANALYSIS

To validate the generalizability of our entropy-based complexity assessment within a wider range of difficulty variations, we extended our analysis across three datasets of increasing difficulty: GSM8K (elementary arithmetic), MATH500 (high school mathematics) and AIME24 (competition-level problems). We conducted 2, 8 and 12 samples GSM8K, MATH500 and AIME24 respectively and calculated their respective averages. This cross-dataset analysis provides crucial evidence that entropy patterns consistently reflect problem complexity across diverse mathematical domains.

### A.3.1 DATASET COMPLEXITY HIERARCHY

Our analysis spans three carefully selected datasets representing distinct complexity levels:

- **GSM8K**: Grade school math problems requiring basic arithmetic and simple reasoning
- **MATH500**: High school mathematics covering algebra, geometry and calculus
- **AIME24**: Competition mathematics demanding advanced problem-solving techniques

### A.3.2 ENTROPY DISTRIBUTION EVOLUTION

Figure 13 reveals systematic shifts in entropy distributions as problem complexity increases. For GSM8K, 45% of tokens fall in the low-mid entropy range (0.1-0.5), reflecting the straightforward nature of elementary arithmetic. This proportion decreases to 32% for MATH500 and further to

30% for AIME24. Conversely, high-entropy tokens ($> 1.0$) increase dramatically from 16.0% in GSM8K to 37% in AIME24, representing a 2.3-fold increase.

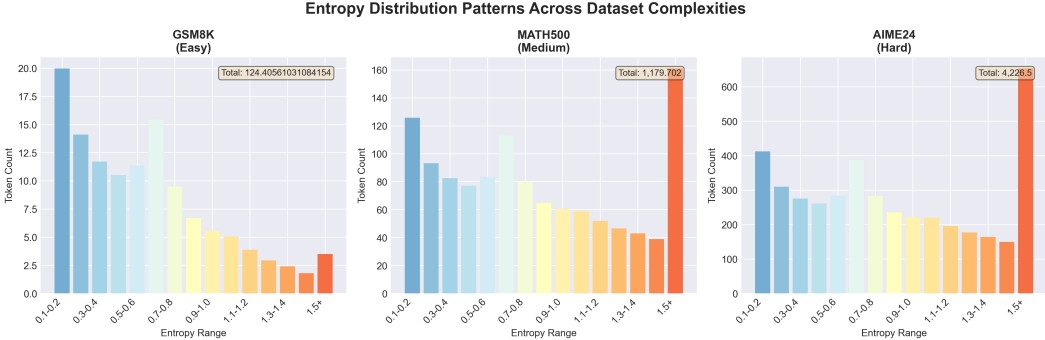

Figure 13: Entropy distribution patterns across dataset complexities.

The heatmap visualization (Figure 14) demonstrates a clear rightward shift in probability mass across entropy ranges. GSM8K exhibits a concentrated distribution in lower entropy bins, while AIME24 shows significant dispersion across the entire entropy spectrum.

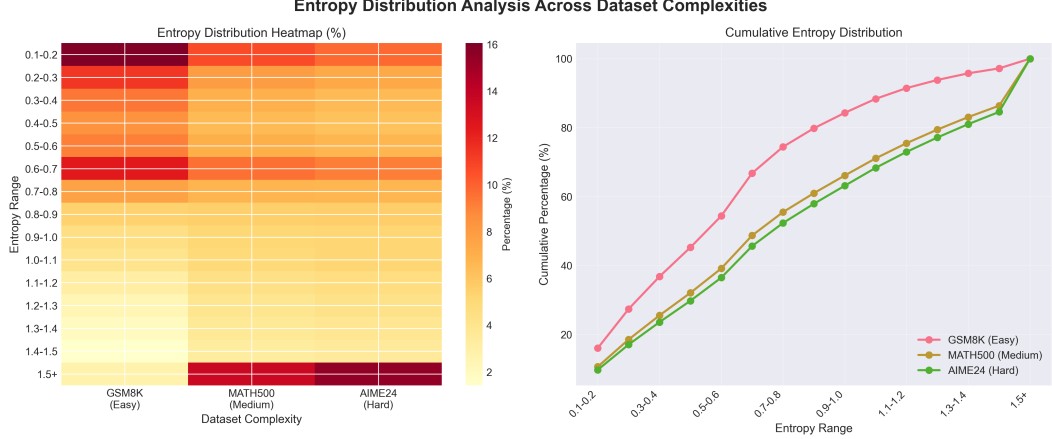

Figure 14: Entropy distribution heatmap and cumulative trends across datasets.

### A.3.3 QUANTITATIVE COMPLEXITY INDICATORS

Statistical analysis (Figure 15) provides compelling evidence for entropy as a complexity measure:

**Mean Entropy Progression:**

- GSM8K: 0.60 (baseline)
- MATH500: 0.82 (+36.7%)
- AIME24: 0.86 (+43.3%)

This 43.3% increase from GSM8K to AIME24 represents a fundamental shift in reasoning patterns, from deterministic arithmetic operations to exploratory mathematical reasoning.

**Token Volume Scaling:** Total meaningful tokens (entropy $> 0.1$) increases 34.1-fold from GSM8K (124 tokens) to AIME24 (4226 tokens), indicating that complex problems require substantially longer reasoning chains.

**Entropy Segmentation Analysis** (Figure 16):

- Low-mid entropy (0.1-0.5): Decreases from 45% to 30%

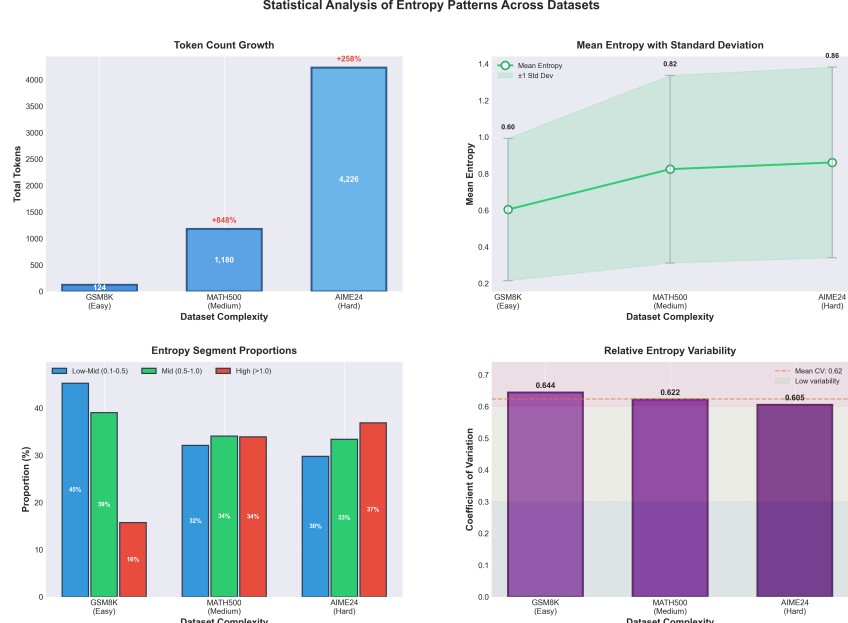

Figure 15: Statistical analysis of entropy patterns across datasets

- Mid entropy (0.5-1.0): Remains relatively stable (39% to 33%)
- High entropy ($> 1.0$): Increases from 16% to 37%

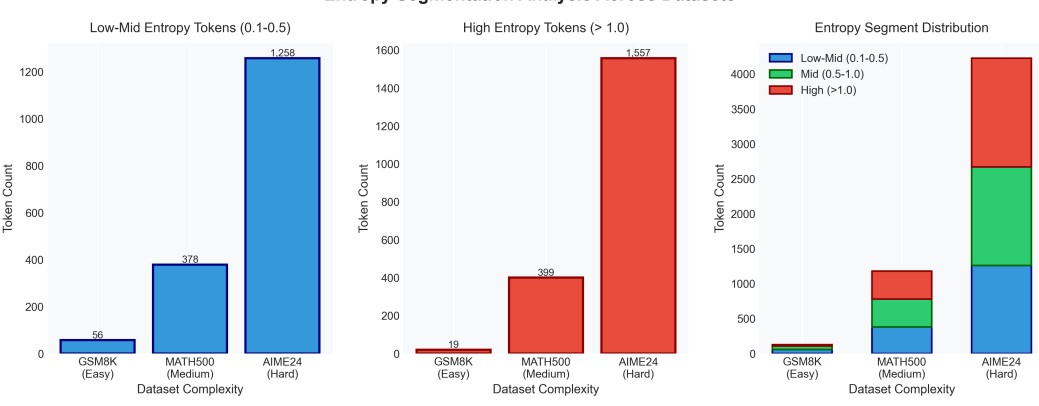

Figure 16: Entropy segmentation analysis across datasets.

The stacked visualization clearly shows the progressive dominance of high-entropy tokens in complex datasets, with AIME24 containing 4226 high-entropy tokens compared to just 124 in GSM8K.

### A.3.4 TOKEN-LEVEL BEHAVIORAL CONSISTENCY

Analysis of high-frequency tokens (Figure 17) reveals remarkable stability in individual token entropy despite dramatic frequency changes. Core reasoning tokens like "the", "I" and "that" maintain entropy values between 1.72-1.87 across all datasets, while their frequency increases exponentially (up to 100-fold for certain tokens).

This stability-frequency paradox indicates that:

1. Model uncertainty for individual reasoning decisions remains consistent

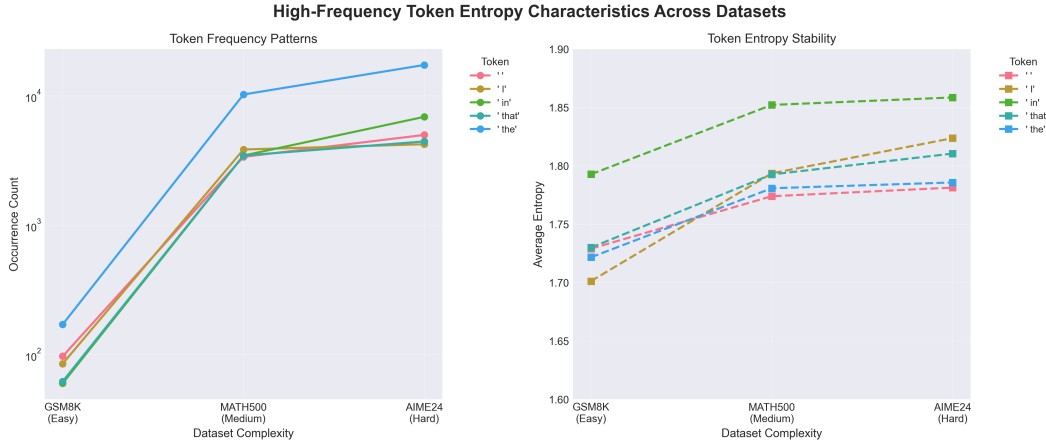

Figure 17: High-frequency token entropy characteristics across datasets.

2. Complex problems require more frequent decision points rather than higher per-decision uncertainty

3. The entropy increase emerges from aggregate effects rather than fundamental changes in token-level behavior

### A.3.5 Implications for Adaptive Reasoning

The cross-dataset analysis provides strong validation for entropy-based adaptive reasoning:

**Robust Complexity Signal**: The monotonic increase in mean entropy ($0.60 \rightarrow 0.86$) across datasets of known difficulty confirms entropy as a reliable complexity indicator independent of specific mathematical domains.

**Clear Decision Boundaries**: The consistent segmentation at entropy thresholds 0.5 and 1.0 across all datasets suggests natural breakpoints for adaptive reasoning strategies:

- Entropy $< 0.5$: Direct response appropriate
- Entropy $0.5 - 1.0$: Moderate reasoning required
- Entropy $> 1.0$: Full deliberative reasoning necessary

**Scalability**: The $34.1\times$ token increase coupled with maintained entropy patterns demonstrates that our approach scales effectively from simple arithmetic to competition-level mathematics.

### A.3.6 Coefficient of Variation Analysis

The coefficient of variation (standard deviation/mean) shows interesting dynamics:

- GSM8K: 0.644 (high relative variability)
- MATH500: 0.622 (moderate decrease)
- AIME24: 0.605 (lowest relative variability)

Counterintuitively, while absolute entropy increases with complexity, relative variability decreases. This suggests that complex problems exhibit more consistent uncertainty patterns, possibly due to the systematic application of advanced problem-solving heuristics.

### A.3.7 Conclusion

The cross-dataset analysis conclusively demonstrates that entropy serves as a domain-invariant measure of mathematical reasoning complexity. The systematic progression from GSM8K through MATH500 to AIME24 validates our theoretical framework and provides empirical support for

entropy-driven adaptive reasoning. The 43.3% increase in mean entropy, coupled with the 2.3-fold increase in high-entropy token proportion, establishes clear quantitative thresholds for reasoning mode selection that generalize across mathematical domains of varying complexity.

# B EXPERIMENT SETUP

## B.1 DATASETS

The dataset used for training is MATH (Hendrycks et al., 2021). It comprises approximately 10K math problem of high school level accompanied with both ground truth solution and ground truth answer. Since the ground truth solution is not need for our experiment, we only use the problem-answer pairs. For training, we selected 5,000 problems from the MATH Train set. We generated 16 solutions for each problem. The dataset utilized for testing encompasses the test sets of MATH500 (Lightman et al., 2023), AIME24 and GPQA (Rein et al., 2024). The following is a brief introduction to these dataset.

- **MATH500** is a subset of 500 problems sampled from the MATH dataset, covering algebra, calculus, geometry and number theory. This dataset is designed to test mathematical reasoning across a range of difficulty levels.
- **AIME24** dataset contains problems from the American Invitational Mathematics Examination (AIME) 2024 and is a prestigious high school mathematics competition known for its challenging mathematical problems.
- **GPQA** comprises graduate-level STEM reasoning tasks. The GPQA Diamond dataset is a benchmark dataset focused on multi-domain knowledge question-answering, constructed by a professional research team.

## B.2 GENERATION CONFIGURATION

We inference on DeepSeek-R1-Distill models with the following generation parameters:

- **Temperature**: 0.6
- **Top-p**: 0.95
- **Maximum sequence length**: 32,768 tokens

## B.3 NOTHINKING PROMPT

We employ prompt injection to enforce that the model does not to generate explicit reasoning contents. This approach proves effective for SFT implementation with the following prompt:

```
Okay, I think I have finished reasoning.\n</think>\n\n
```

## B.4 ENTROPY-BASED COMPLEXITY ASSESSMENT

Based on our experiments in Section 3, we identify the optimal entropy threshold of 0.2 for the MATH500 dataset, which yields the strongest correlation between average entropy and problem difficulty. Our entropy calculation involves two key steps: first, we compute token-level entropy using the top-10 token probabilities (renormalized within this subset); second, we apply the 0.2 filtering threshold when averaging across the generated sequence.

## B.5 TRAINING CONFIGURATION

### B.5.1 STAGE 1: SUPERVISED FINE-TUNING (SFT)

The SFT stage employs the following training parameters for NoThinking exemplars:

- **Learning rate**: 5.0e-6 (may vary from model sizes)

- **Number of epochs**: 3.0 (may vary from model sizes)
- **Cutoff length**: 8,192 tokens
- **Training samples**: 1500 NoThinking exemplars
- **Training duration**: 10 minutes on 2 A100 80GB GPUs

### B.5.2  STAGE 2: OFFLINE PROXIMAL POLICY OPTIMIZATION (PPO)

The PPO stage optimizes reasoning mode selection through entropy-driven reward mechanisms:

- **Learning rate**: 5.0e-7 (may vary from model sizes)
- **Number of epochs**: 1.0
- **Cutoff length**: 12,288 tokens
- **Training samples**: 10,000 problem instances
- **Training duration**: 1.5 hours on 4 A100 80GB GPUs
- **Reward weights**: $w_1$=0.3 (accuracy), $w_2$=0.5 (efficiency penalty), $w_3$=0.3 (difficulty-adaptive)
- **Candidates per question (K)**: 16 (offline dataset and baseline reward averaging)

Our training configuration achieves an efficient balance between computational cost and model performance, enabling practical deployment in resource-constrained environments.

## C  COMPLEMENTARY EXPERIMENTS

### C.0.1  ABLATION STUDY

To validate the contribution of each component within our framework, we conducted a series of ablation studies, with the results detailed in Table 2. The findings clearly indicate that both the SFT and PPO stages are crucial for achieving the final performance.

First, removing the PPO stage (w/o PPO) resulted in a notable performance degradation across all models. For the DeepSeek-1.5B model, relying solely on the SFT-only method led to a 6.2% decrease in average accuracy and a 36.9% increase in response length. This demonstrates that the PPO stage is indispensable for refining the model's policy, further boosting accuracy and generating more concise answers.

Second, we evaluated the importance of the difficulty-adaptive reward (also referred to as confidence reward, in Formula 7) within the PPO stage. Upon removing this reward component (w/o Confidence Reward), the models experienced an even more significant drop in average accuracy, with the DeepSeek-1.5B model showing a 6.2% decrease. This highlights the critical role of our entropy-based difficulty-adaptive reward in guiding the model to learn when to engage in deep reasoning versus when to provide a direct answer.

In summary, the results of the ablation study confirm the integrity of our two-stage framework and the necessity of each component. The SFT stage provides the model with foundational capability, while the PPO stage, guided by the difficulty-adaptive reward, further refines and enhances its performance. The two stages are complementary and work in synergy to achieve superior final results.

### C.0.2  ENTROPY THRESHOLD SENSITIVITY ANALYSIS

To investigate the robustness of our entropy-based complexity assessment, we conducted a comprehensive analysis of the entropy threshold hyperparameter $\tau$ using the DeepSeek-8B model. We evaluated thresholds ranging from 0.0 to 0.5 across all three benchmarks, measuring the impact on accuracy and response length.

Table 3 shows the performance variation across different entropy thresholds. From the experimental results, $\tau = 0.2$ demonstrates relatively good performance across multiple datasets, achieving optimal results on AIME24 and GPQA while maintaining competitive accuracy on MATH500. The threshold also leads to effective response length reduction compared to lower values. However, further analysis would be beneficial to fully understand the underlying mechanisms and optimize threshold selection for different task types.

Table 2: Ablation study.

| Method | MATH 500 | | AIME 2024 | | GPQA | | Average | |
|---|---|---|---|---|---|---|---|---|
| | Acc | Length | Acc | Length | Acc | Length | ΔAcc | ΔLength |
| *DeepSeek-R1-Distill-Qwen-1.5B* | | | | | | | | |
| Full (SFT+PPO) | **84.2** | 3257 | **33.3** | **7250** | **39.4** | 5260 | - | - |
| w/o PPO (SFT-only) | 80.2 | 4116 | 26.7 | 10821 | 31.3 | 7114 | -6.2 | +36.9% |
| w/o Confidence Reward | 80.8 | **2765** | 26.7 | 8489 | 30.8 | 5522 | -6.2 | +2.3% |
| *DeepSeek-R1-Distill-Qwen-7B* | | | | | | | | |
| Full (SFT+PPO) | **92.4** | 3105 | **63.3** | 10230 | **49.0** | 6626 | - | - |
| w/o PPO (SFT-only) | 91.2 | 3259 | 53.3 | 12156 | 48.5 | **6516** | -3.9 | +7.4% |
| w/o Confidence Reward | 88.6 | **2807** | 53.3 | 11992 | 45.5 | 6588 | -5.8 | +2.4% |
| *DeepSeek-R1-Distill-Llama-8B* | | | | | | | | |
| Full (SFT+PPO) | 85.8 | **3564** | **56.7** | **10126** | **49.0** | **7135** | - | - |
| w/o PPO (SFT-only) | **88.0** | 4557 | 43.3 | 13155 | 47.0 | 7666 | -4.4 | +21.7% |
| w/o Confidence Reward | 86.0 | 3582 | 43.3 | 11146 | 43.9 | 7749 | -6.1 | +6.4% |

Table 3: Performance across different entropy thresholds $\tau$ on DeepSeek-R1-Distill-Llama-8B.

| $\tau$ | MATH500 | | AIME24 | | GPQA | |
|---|---|---|---|---|---|---|
| | Acc | Length | Acc | Length | Acc | Length |
| 0.0 | 85.4 | 4139 | 40.0 | 11071 | 44.4 | 8260 |
| 0.01 | 85.6 | 4117 | 40.0 | 12724 | 48.0 | 8439 |
| 0.1 | 86.6 | 3705 | 50.0 | 12360 | 47.5 | 7894 |
| 0.2 | 85.8 | 3564 | **56.7** | **10126** | **49.0** | **7135** |
| 0.5 | **87.6** | **3461** | 40.0 | 11380 | 45.0 | 7757 |

### C.0.3    GENERALIZATION ON MMLU DATASET

To assess the generalization capabilities of our approach, we evaluated its performance on the MMLU dataset, an out-of-distribution benchmark. As shown in Table 4, our method demonstrates strong generalization across model scales. For Deepseek-1.5B, 7B and 8B models, our approach maintains nearly unchanged accuracy compared to the original thinking models while achieving substantial reductions in response length. This suggests that the learned efficiency and reasoning optimization strategies can be effectively transferred to new, unseen domains, demonstrating the robustness and scalability of our framework.

## D    CASE STUDY

We select representative problems based on MATH500's difficulty ratings: the "easy" problem has a difficulty score of 2.0, while the "hard" problem has a difficulty score of 5.0. Figure 18 presents the prompt template and problem instance for the easy case, while Figure 19 shows the corresponding template and instance for the hard case.

The comparative analysis demonstrates how our PPO-trained models learn to adaptively select appropriate reasoning modes. For the easy problem, Figures 20 and 21 show how both 1.5B and 8B models learn to use NoThinking mode, generating concise solutions without unnecessary deliberation. For the hard problem, Figures 22 and 23 illustrate how both models appropriately engage Thinking mode, employing extensive reasoning to solve complex problems. This adaptive behavior showcases the effectiveness of our training approach in teaching models when to think deeply versus when to respond directly.

Table 4: The performance on out-of-distribution test set MMLU.

| Method | MMLU | | | |
|---|---|---|---|---|
| | Acc | Length | ΔAcc | ΔLength |
| *DeepSeek-R1-Distill-Qwen-1.5B* | | | | |
| Original$_{\text{Thinking}}$ | 39.4 | 2241 | - | - |
| Ours | **41.7** | **778** | +2.3 | -65.3% |
| *DeepSeek-R1-Distill-Qwen-7B* | | | | |
| Original$_{\text{Thinking}}$ | **63.7** | 1462 | - | - |
| Ours | 61.5 | **881** | -2.2 | -39.7% |
| *DeepSeek-R1-Distill-Llama-8B* | | | | |
| Original$_{\text{Thinking}}$ | 69.7 | 1891 | - | - |
| Ours | **70.2** | **1461** | +0.5 | -22.7% |

---

**Prompt Template for Easy Problem**

**System Prompt:**
You are a helpful assistant. You should think step-by-step and put your final answer within `\\boxed{{}}`.

---

**Problem:**
Solve the problem: In a class of 50 students, 28 participate in MATHCOUNTS, 21 participate in science club, and 6 students participate in neither. How many students participate in both MATHCOUNTS and science club? `<think>`

Figure 18: Prompt template for relatively easy questions from MATH500, difficulty 2.0

---

**Prompt Template for Hard Problem**

**System Prompt:**
You are a helpful assistant. You should think step-by-step and put your final answer within `\\boxed{{}}`.

---

**Problem:**
Solve the problem: Six cars pull up to a red light, one at a time. At the light, there are three lanes, one left-turn lane, one straight-going lane, and one right-turn lane. How many ways can the cars stack up so that all three lanes are occupied? Note that if the first car turns left and the second goes straight, this is considered different from the first car going straight and the second car turning left. In other words, the cars are distinguishable, but pull up to the intersection in a fixed order. `<think>`

Figure 19: Prompt template for hard questions from MATH500, difficulty 5.0

**DeepSeek-R1-Distill-Qwen-1.5B**

**Before PPO:**

Okay, so I have this problem here: There are 50 students in a class. Out of these, 28 are in MATHCOUNTS, 21 are in the science club, and 6 are in neither. I need to find out how many students are in both MATHCOUNTS and the science club. Hmm, let me think. This seems like a problem involving sets and maybe the principle of inclusion-exclusion. I remember that when dealing with two overlapping sets, the total number of elements in either set is equal to the sum of the elements in each set minus the elements in their intersection. So, if I denote the number of students in both MATHCOUNTS and science club as $x$, then I can set up an equation. Let me write that down. The total number of students is 50. But 6 are in neither club, so the number of students in at least one of the clubs is $50 - 6 = 44$. That makes sense because 6 students are excluded from both. Now, according to the inclusion-exclusion principle, the number of students in MATHCOUNTS plus the number of students in the science club minus the number of students in both equals the number of students in at least one club. So, mathematically, that would be: $|M| + |S| - |M \cap S| =$ Total in at least one club Plugging in the numbers, that's: 28 (MATHCOUNTS) + 21 (science club) $- x$ (both) $= 44$ So, $28 + 21 - x = 44$ Adding 28 and 21 together gives 49, so: $49 - x = 44$ To find $x$, I subtract 44 from both sides: $49 - 44 = x$ Which simplifies to: $5 = x$ So, $x$ is 5. That means 5 students are in both MATHCOUNTS and the science club. Wait, let me double-check. If 5 students are in both, then the number of students only in MATHCOUNTS would be $28 - 5 = 23$, and only in science club would be $21 - 5 = 16$. Adding those up: $23 + 16 + 5$ (both) $= 44$, which is exactly the number of students in at least one club. Adding the 6 students in neither gives $44 + 6 = 50$, which matches the total number of students in the class. So, that checks out. I think that's correct. There are 5 students who are in both clubs. </think>To solve the problem, we can use the principle of inclusion-exclusion.

1. **Total students**: 50; 2. **Students in neither club**: 6; 3. **Students in at least one club**: 50 - 6 = 44. Let: $|M|$ = number of students in MATHCOUNTS = 28, $|S|$ = number of students in science club = 21, $|M \cap S|$ = number of students in both clubs = x (unknown).

Using the inclusion-exclusion principle:

$$|M \cup S| = |M| + |S| - |M \cap S|$$
$$44 = 28 + 21 - x$$
$$44 = 49 - x$$
$$x = 49 - 44$$
$$x = 5$$

**Answer:** The number of students participating in both MATHCOUNTS and the science club is $\boxed{5}$. **(800 tokens)**

---

**After PPO:**

Okay, I think I have finished reasoning.</think> To solve the problem, we'll use the principle of inclusion-exclusion. 1. **Total Students:** There are 50 students in the class. 2. **Participants in MATHCOUNTS:** 28 students. 3. **Participants in Science Club:** 21 students. 4. **Students in Neither:** 6 students. **Step 1:** Calculate the number of students who are in at least one of the clubs. Total participants in MATHCOUNTS or Science Club = Total students - Students in neither = 50 - 6 = 44 students. **Step 2:** Apply the inclusion-exclusion principle. Number of students in both clubs = Number in MATHCOUNTS + Number in Science Club - Number in at least one club. = 28 + 21 - 44 = 49 - 44 = **5 students**. **(800 tokens)** $\boxed{5}$ students participate in both MATHCOUNTS and the science club. **(219 tokens)**

Figure 20: For this relatively straightforward mathematical problem, DeepSeek-1.5B exhibits unnecessary deliberation and redundant reasoning steps. In contrast, our PPO model adaptively selects NoThinking mode, directly generating a concise solution with only 219 tokens.

**DeepSeek-R1-Distill-Llama-8B**

**Before PPO:**

Okay, so I have this problem here about students participating in MATHCOUNTS and science club. Let me try to figure it out step by step. Hmm, let's see. The class has 50 students in total. Out of these, 28 participate in MATHCOUNTS, 21 participate in science club, and 6 students don't participate in either. I need to find how many students are in both MATH-COUNTS and science club.**[...](omit)** Alternatively, maybe using a Venn diagram. Imagine two overlapping circles, one for MATHCOUNTS with 28 students and one for science club with 21 students. The overlapping part is the number we're trying to find. The total number of students is 50, but 6 are not in either, so 50 - 6 = 44 are in at least one of the clubs. So, the union of both clubs is 44. So, in the Venn diagram, the total area covered by both circles is 44. So, the area covered by MATHCOUNTS alone is 28 - Both, and the area covered by science club alone is 21 - Both. So, adding those gives (28 - Both) + (21 - Both) + Both = 49 - Both. But this sum should equal 44. So, 49 - Both = 44, so Both = 5. Yep, same answer again. So, no matter which way I approach it, I keep getting 5. So, I think that's solid. I don't think I made any mistakes here. Just to recap: Total students = 50 Neither = 6 So, students in at least one club = 50 - 6 = 44 Let B be the number in both. Then, students in only MATHCOUNTS = 28 - B Students in only science club = 21 - B Total in at least one club = (28 - B) + (21 - B) + B = 49 - B But we know that this equals 44. So, 49 - B = 44 Therefore, B = 49 - 44 = 5 Yep, that all lines up. So, I think I can confidently say the answer is 5. **Final Answer** The number of students who participate in both MATHCOUNTS and science club is $\boxed{5}$. </think> Given a class of 50 students, where 28 participate in MATHCOUNTS, 21 participate in science club, and 6 students participate in neither, we need to find the number of students who participate in both MATHCOUNTS and science club. We use the principle of inclusion-exclusion. The formula for the total number of students is:

$$\text{Total} = \text{MATHCOUNTS} + \text{Science Club} - \text{Both} + \text{Neither}$$

Plugging in the given values:
$$50 = 28 + 21 - B + 6$$

Simplifying the equation:
$$50 = 55 - B$$

Solving for $B$:
$$B = 55 - 50 = 5$$

Thus, the number of students who participate in both MATHCOUNTS and science club is $\boxed{5}$. **(1569 tokens)**

---

**After PPO:**

Okay, I think I have finished reasoning. </think> To solve the problem, we can use the principle of inclusion-exclusion. Let's denote: - Total number of students in the class = 50 - Number of students participating in MATHCOUNTS = 28 - Number of students participating in science club = 21 - Number of students participating in neither MATHCOUNTS nor science club = 6 We need to find the number of students who participate in both MATH-COUNTS and science club. First, calculate the total number of students who participate in at least one of the two clubs: Total students in either MATHCOUNTS or science club = Total students - Students in neither = 50 - 6 = 44 Let $x$ be the number of students who participate in both MATHCOUNTS and science club. According to the inclusion-exclusion principle: Number of students in MATHCOUNTS + Number of students in science club = Total students in either club + Number of students in both clubs So, 28 + 21 = 44 + x Solving for $x$: 49 = 44 + x x = 49 - 44 x = 5 Therefore, the number of students who participate in both MATHCOUNTS and science club is $\boxed{5}$. **(284 tokens)**

Figure 21: DeepSeek-8B can see a significant redudancy in tokens after selecting NoThinking mode.

---

**DeepSeek-R1-Distill-Qwen-1.5B**

**Before PPO:**

Okay, I think I have finished reasoning. </think>To solve the problem of determining the number of ways six distinguishable cars can stack up into three lanes (left-turn, straight-going, and right-turn) such that all three lanes are occupied, we can use the concept of permutations. 1. **Understanding the Problem**: - There are six cars, each distinguishable. - They pull in one at a time in a fixed order. - The cars must occupy all three lanes, meaning each lane must have at least one car. 2. **Total Permutations**: - The total number of ways to arrange six cars in order is 6! (6 factorial). - 6! = 720. 3. **Subtracting Unoccupied Cases**: - We need to subtract the cases where one or more lanes are empty. - This can be calculated using the principle of inclusion-exclusion. 4. **Applying Inclusion-Exclusion Principle**: - Let $A$, $B$, and $C$ represent the cases where the left-turn, straight-going, and right-turn lanes are empty, respectively. - The number of ways where at least one lane is empty is $|A| + |B| + |C| - |A \cap B| - |A \cap C| - |B \cap C| + |A \cap B \cap C|$. 5. **Calculating Each Term**: - $|A| = |B| = |C| = 5! = 120$. - $|A \cap B| = |A \cap C| = |B \cap C| = 4! = 24$. - $|A \cap B \cap C| = 3! = 6$. 6. **Putting It All Together**: - Total ways with all lanes occupied: $6! - 3 \times 5! + 3 \times 4! - 3! = 720 - 360 + 72 - 6 = 426$. **Final Answer**: The number of ways the cars can stack up so that all three lanes are occupied is $\boxed{426}$. **(483 tokens)**

---

**After PPO:**

Okay, I need to solve this problem about six cars pulling up to a red light with three lanes: left-turn, straight-going, and right-turn. All three lanes must be occupied, and the cars are distinguishable but pull up one by one in a fixed order. Hmm, let me think about how to approach this. First, I need to understand the problem clearly. There are six cars, each pulling up one after another. The order in which they pull up is fixed—they are distinguishable, so each car is unique. But since they pull up in a fixed order, I think the order is already determined and doesn't change? Wait, maybe I need to clarify that. **[...](omit)** Therefore, I think the correct answer is 540. **Final Answer** The number of ways is $\boxed{540}$. </think>
To solve the problem of determining how many ways six cars can stack up to a red light with three lanes such that all lanes are occupied, we need to consider the distribution of cars into the three lanes. Each lane must have at least one car. We use the principle of inclusion-exclusion to count the number of onto functions from six cars to three lanes. The formula for the number of onto functions is:

$$\sum_{i=0}^{k}(-1)^i \binom{k}{i}(k-i)^n$$

where $n = 6$ (the number of cars) and $k = 3$ (the number of lanes). Applying the formula:

$$3^6 - 3 \cdot 2^6 + 3 \cdot 1^6$$

Calculating each term:

$$3^6 = 729$$
$$3 \cdot 2^6 = 3 \cdot 64 = 192$$
$$3 \cdot 1^6 = 3 \cdot 1 = 3$$

Putting it all together:

$$729 - 192 + 3 = 540$$

Thus, the number of ways the cars can stack up so that all three lanes are occupied is $\boxed{540}$.
**(2438 tokens)**

---

Figure 22: DeepSeek-1.5B can determine hard problems using Thinking mode.

**DeepSeek-R1-Distill-Llama-8B**

**Before PPO:**
Okay, I think I have finished reasoning. `</think>` To solve the problem, let's break it down step by step. 1. **Understand the Problem:** - There are six cars that pull up to a red light one by one. - The light has three lanes: left-turn, straight-going, and right-turn. - Each car must choose one of the three lanes. - The order of the cars matters because they are distinguishable. 2. **Determine the Total Number of Possibilities:** - Each car has 3 choices: left, straight, or right. - Since the cars are distinguishable and choose lanes in sequence, this is a permutation problem with repetition allowed. - The total number of ways the cars can choose lanes is $3^6$. 3. **Calculate the Total Number of Ways:** - $3^6 = 3 \times 3 \times 3 \times 3 \times 3 \times 3 = 729$. 4. **Conclusion:** - Therefore, there are $\boxed{729}$ ways the cars can stack up so that all three lanes are occupied. **(245 tokens)**

---

**After PPO:**
Okay, so I have this problem here where six cars pull up to a red light one at a time. There are three lanes: left-turn, straight, and right-turn. I need to figure out how many ways the cars can stack up so that all three lanes are occupied. Hmm, let me break this down step by step. First, I think it's about permutations because the order matters here. The problem says that if the first car turns left and the second goes straight, that's different from the first going straight and the second turning left. So each car is distinguishable, but they all arrive in a fixed order. So, the order in which they arrive matters. **[...](omit)** So yes, the number is $3^6 - 3*2^6 + 3*1^6 = 729 - 192 + 3 = 540$. So, yeah, the final answer is 540. **Final Answer** The number of ways is $\boxed{540}$. `</think>` To solve the problem of determining how many ways six cars can pull up to a red light such that all three lanes (left-turn, straight, and right-turn) are occupied, we need to consider the order of the cars and the lanes they choose. Each car is distinguishable and arrives one at a time, and the order matters. We can use the principle of inclusion-exclusion to count the number of valid sequences where each lane is used at least once. The total number of possible sequences without any restrictions is $3^6$. We then subtract the cases where one or more lanes are empty and add back the cases where two lanes are empty. 1. **Total number of sequences**: $3^6 = 729$ 2. **Subtract sequences where one lane is empty**: - Each lane being empty reduces the number of choices to 2 for each car, so we subtract $3 \times 2^6 = 3 \times 64 = 192$. 3. **Add back sequences where two lanes are empty**: - Each case where two lanes are empty reduces the number of choices to 1 for each car, so we add back $3 \times 1^6 = 3 \times 1 = 3$. Putting it all together:

$$3^6 - 3 \times 2^6 + 3 \times 1^6 = 729 - 192 + 3 = 540$$

Thus, the number of ways the cars can stack up so that all three lanes are occupied is $\boxed{540}$. **(2910 tokens)**

Figure 23: DeepSeek-8B can determine hard problems choosing Thinking mode.

