# OpenReview forum: "Discovering and Leveraging Entropy-Complexity Relationships for Efficient Large Language Model Reasoning"
_ICLR.cc/2026/Conference — Submitted to ICLR 2026_

### Official Review · Reviewer_1kVB · 2025-10-30

**Soundness:** 1
**Presentation:** 1
**Contribution:** 1
**Rating:** 0
**Confidence:** 5

**Summary:**

This paper focuses on reducing the overthinking problem in LLMs. It shows that token-level entropy correlates with problem complexity, allowing models to decide when to answer directly or perform full reasoning. The method includes a two-stage training process: supervised fine-tuning on simple “NoThinking” samples, followed by offline PPO with an entropy-aware reward.

**Strengths:**

The experimental details in this paper are thorough, and experiments have been conducted on multiple models.

**Weaknesses:**

1. The correlation with difficulty in Figure 2 is not clearly explained. Although it is presented as a key motivation, there is no detailed discussion in the paper. It would be helpful to provide more context and clarification regarding this metric.
2. The paper identifies the problem of excessive reasoning in large models, where long reasoning chains reduce computational efficiency. However, the entropy-based approach discussed in the paper does not seem directly related to this issue. The paper seems to address this problem by imposing an **Efficiency Penalty** on reasoning length. This aspect feels somewhat inconsistent with the earlier points made in the article.
3. The optimization algorithm presented in the second phase differs significantly from PPO. I did not observe the calculation of advantage functions that adhere to the Bellman equation, which makes the method appear more like GRPO. Referring to it as PPO might be misleading.
4. It would be beneficial for the paper to include a more thorough discussion of related works, such as [1, 2, 3].
5. I recommend adding more ablation studies to the paper, particularly experiments evaluating the impact of different weights like $w_1$, $w_2$, $w_3$, and the influence of each component in the reward function on the results.
6. The paper lacks comparisons with related experiments, such as [4, 5]. Including such comparisons would strengthen the paper's contributions.
7. A significant concern is that the relationship between problem difficulty, entropy, and model overthinking is not sufficiently clarified. While some correlation experiments are presented, they are not comprehensive enough. It would be valuable to see more detailed analyses or theoretical proofs addressing these interconnections.



[1] The Entropy Mechanism of Reinforcement Learning for Reasoning Language Models

[2] Beyond the 80/20 Rule: High-Entropy Minority Tokens Drive Effective Reinforcement Learning for LLM Reasoning

[3] REASONING WITH EXPLORATION: AN ENTROPY PER-SPECTIVE ON REINFORCEMENT LEARNING FOR LLMS

[4] Token-Budget-Aware LLM Reasoning

[5] CoT-Valve: Length-Compressible Chain-of-Thought Tuning

**Questions:**

Please refer to Weaknesses.

---

> ### Author Response · Authors · 2025-11-18
> **Official Response to Reviewer 1kVB Part (1/3)**
>
> ### Reply to Weakness 1: Explanation of Problem Difficulty Correlation
>
> > The correlation with difficulty in Figure 2 is not clearly explained. Although it is presented as a key motivation, there is no detailed discussion in the paper. It would be helpful to provide more context and clarification regarding this metric.
>
> Thank you for your valuable question. Regarding the definition of problem difficulty: we reference the manually annotated difficulty coefficients in the MATH dataset and the source of problems. From the perspective of large model performance, we consider that problem difficulty represents the necessary computational complexity during the model's problem-solving process, as well as the model's uncertainty level in responses. These are reflected in specific metrics such as response length and accuracy, which we present in Appendix A.1 of the paper, discussing model performance under human-like thinking difficulty levels. We will further elaborate and clarify this definition in the revised version.
>
> ### Reply to Weakness 2: Efficiency Penalty vs. Length Penalty
>
> > The paper identifies the problem of excessive reasoning in large models, where long reasoning chains reduce computational efficiency. However, the entropy-based approach discussed in the paper does not seem directly related to this issue. The paper seems to address this problem by imposing an Efficiency Penalty on reasoning length. This aspect feels somewhat inconsistent with the earlier points made in the article.
>
> Thank you for your valuable question. Our task is not simply to achieve shortening of the reasoning process, but more importantly, to maintain necessary reasoning complexity while shortening unnecessary length (the difficulty-aware reward component achieves this, which includes average entropy and has a direct relationship with problems). Reasoning efficiency penalty is only one aspect of this training objective. Considering length penalty alone cannot solve the problem of maintaining necessary complexity.
>
> ### Reply to Weakness 3: PPO vs GRPO
>
> > The optimization algorithm presented in the second phase differs significantly from PPO. I did not observe the calculation of advantage functions that adhere to the Bellman equation, which makes the method appear more like GRPO. Referring to it as PPO might be misleading.
>
> Thank you for your correction. We use the term PPO primarily to indicate the use of its importance sampling and clipping mechanisms. We also note that this processing approach is more similar to GRPO's approach. In the revised version, we will consider revising the wording for clarification.
>
> ### Reply to Weakness 4: Discussion of Related Work
>
> > It would be beneficial for the paper to include a more thorough discussion of related works, such as [1, 2, 3].
>
> Work [1] primarily studies the entropy collapse mechanism during RL training of large models and its fit with model performance, exploring the reasons for entropy value changes during training. Our work focuses on exploring the relationship between the distribution of basic token entropy in thinking model outputs and problem difficulty as well as response cost. Although both utilize entropy values in models, the task scenarios and methods are completely different. Work [2] is one of the foundations of our method. We used its conclusion that mathematical reasoning problems contain a large number of low-entropy tokens with low contribution, which has been discussed in the paper. Work [3] points out the important role of high-entropy tokens in exploration and encourages model exploration by amplifying related token rewards. This has certain relevance to our work, but it is worth noting that the focus of our work is exploring the distribution of token entropy values in thinking models, which is not reflected in related work, and the goal of encouraging model exploration also differs from our goal of deciding whether to encourage exploration based on question difficulty. In the revised version, we will introduce discussions of the other two works, but it is worth noting that entropy is widely used in LLMs, with numerous related papers, and we can only analyze and discuss entropy utilization versions that are more similar to our work, making comprehensive coverage difficult.

---

> ### Author Response · Authors · 2025-11-18
> **Official Response to Reviewer 1kVB Part (2/3)**
>
> ### Reply to Weakness 5: Ablation of Reward Components
>
> > I recommend adding more ablation studies to the paper, particularly experiments evaluating the impact of different weights like $w_1$, $w_2$, $w_3$, and the influence of each component in the reward function on the results.
>
> In the ablation experiments in Appendix C, we ablated the effectiveness of our proposed difficulty-aware reward component, and experimental results demonstrated the importance of our proposed difficulty-aware reward for the model. The influence of other components of the reward function has been discussed in baseline work.
>
> Due to unavailability of previous training resources, we switched to different training GPU resources, which led to partial inconsistencies between optimal hyperparameters and results compared to the original manuscript. However, the overall performance differences are minimal. We have listed the differences (from left to right, Acc and Length correspond to three datasets: MATH500, AIME2024, GPQA, aligned with columns in Table 1 of the paper. The hyperparameters correspond to correctness reward coefficient, length penalty coefficient, and difficulty-adaptive coefficient. For readability, we use one decimal place for distinction).
>
> | Hyperparameters | MATH-Acc↑ | MATH-Length↓ | AIME-Acc↑ | AIME-Length↓ | GPQA-Acc↑ | GPQA-Length↓ |
> | --------------- | --------- | ------------ | --------- | ------------ | --------- | ------------ |
> | origin          | 92.4      | 3105         | 63.3      | 10230        | 49.0      | 6626         |
> | 1-5-3           | 89.2      | 3315         | 63.3      | 10383        | 50.0      | 6824         |
> | 3-3-3           | 92.2      | 3205         | 56.7      | 11793        | 46.0      | 6773         |
> | 3-5-1           | 91.0      | 3331         | 53.3      | 13643        | 49.0      | 6655         |
> | 3-5-3           | 91.6      | 3254         | 56.7      | 12002        | 52.5      | 6467         |
> | 3-5-5           | 90.4      | 3108         | 56.7      | 12570        | 54.0      | 6679         |
> | 3-7-3           | 90.6      | 3128         | 60.0      | 10896        | 48.5      | 6536         |
> | 5-3-3           | 90.0      | 3240         | 53.3      | 10863        | 47.5      | 6794         |
> | 5-3-5           | 92.2      | 3380         | 53.3      | 9458         | 49.0      | 6676         |
> | 5-5-3 *         | 91.4      | 3148         | 63.3      | 9815         | 49.0      | 6830         |
> | 5-5-5           | 89.6      | 3268         | 46.7      | 12279        | 48.5      | 6461         |
> | 7-5-3           | 90.2      | 3252         | 63.3      | 12763        | 53.0      | 6612         |
>
> Based on the experimental results, the three hyperparameters exhibit different impact patterns on model performance:
>
> The influence of the correctness reward coefficient ($w_1$) presents nonlinear characteristics. For in-domain datasets, medium-low values ($w_1$=0.3-0.5) can achieve optimal accuracy (around 92%), but for out-of-domain high-difficulty tasks, higher correctness rewards ($w_1$=0.7) show advantages, particularly achieving 63.3% accuracy on the AIME dataset. This indicates that the role of correctness reward is task-dependent—simple tasks do not require overly strong correctness incentives, while complex reasoning tasks benefit from stronger correctness guidance.
>
> The length penalty coefficient ($w_2$) demonstrates clear and stable control over output efficiency. When $w_2$=0.5, the model produces more concise outputs across all datasets. For example, on the AIME dataset, configurations with $w_2$=0.5 average around 10,000 words, significantly shorter than configurations with $w_2$=0.3. Interestingly, moderate length penalty ($w_2$=0.5) does not harm accuracy; rather, it improves out-of-domain generalization in some configurations, possibly because enforcing concise expression compels the model to learn more essential reasoning patterns.
>
> The difficulty-adaptive coefficient ($w_3$) exhibits interactions with task difficulty. Medium values ($w_3$=0.3) perform best on the high-difficulty AIME dataset (multiple configurations achieve 63.3%), while higher values ($w_3$=0.5) show slight advantages on the medium-difficulty GPQA. This "counter-intuitive" phenomenon suggests that excessive difficulty adaptation may lead the model to over-adjust strategies on truly difficult tasks, while moderate adaptation ($w_3$=0.3) provides more robust performance.
>
> There exists synergy among the three hyperparameters: the balanced configuration of 5-5-3 exhibits the best comprehensive performance, indicating that in reinforcement learning fine-tuning, moderate balance across dimensions is more important than extreme optimization of any single dimension. This finding has important practical implications—pursuing overall balanced reward design cultivates model robustness and generalization capability better than emphasizing any single aspect.

---

> ### Author Response · Authors · 2025-11-18
> **Official Response to Reviewer 1kVB Part (3/3)**
>
> ### Reply to Weakness 6: More Baseline Comparisons
>
> > The paper lacks comparisons with related experiments, such as [4, 5]. Including such comparisons would strengthen the paper's contributions.
>
> Thank you for your valuable suggestion. Due to limited computational resources or lack of open-source code, we were unable to directly reproduce AdaptThink (requiring 32×A100) and other methods requiring substantial computational resources. O1-Pruner was the only comparison method we found that uses the same offline reinforcement learning training process. However, since AdaptThink provides open-source models, we present comparison experimental results based on their open-source models AdaptThink-1.5B-delta0.05 and AdaptThink-7B-delta0.05 (see tables above). We would also like to note that the relatively lower training cost is one of the advantages of our method.
>
> **DeepSeek-R1-Distill-Qwen-1.5B**
>
> | Method              | MATH-Acc↑ | MATH-Length↓ | AIME-Acc↑ | AIME-Length↓ | GPQA-Acc↑ | GPQA-Length↓ | ΔAcc↑    | ΔLength↓   |
> | ------------------- | --------- | ------------ | --------- | ------------ | --------- | ------------ | -------- | ---------- |
> | Original_Thinking   | 85.2      | 5461         | 23.3      | 17231        | 39.4      | 9405         | -        | -          |
> | Original_NoThinking | 70.4      | 944          | 10.0      | 2635         | 13.6      | 782          | -18.0    | -86.4%     |
> | Fast-Solving Prompt | 78.2      | 3344         | 26.7      | 14424        | 36.9      | 9312         | -2.0     | -18.7%     |
> | SFT                 | 80.6      | 4965         | 23.3      | 17575        | 32.8      | 10322        | -3.7     | +0.9%      |
> | DPO                 | **84.8**  | 4816         | **33.3**  | 14219        | 34.8      | 9556         | +1.7     | -9.2%      |
> | O1-Pruner           | 82.8      | 4635         | 30.0      | 16841        | 37.9      | 8869         | +0.9     | -7.7%      |
> | AdaptThink          | 84.4      | **1985**     | 30.0      | 9192         | 35.3      | 7854         | +0.6     | -42.3%     |
> | Ours                | 84.2      | 3257         | **33.3**  | **7250**     | **39.4**  | **5260**     | **+3.0** | **-47.5%** |
>
> **DeepSeek-R1-Distill-Qwen-7B**
>
> | Method              | MATH-Acc↑ | MATH-Length↓ | AIME-Acc↑ | AIME-Length↓ | GPQA-Acc↑ | GPQA-Length↓ | ΔAcc↑    | ΔLength↓   |
> | ------------------- | --------- | ------------ | --------- | ------------ | --------- | ------------ | -------- | ---------- |
> | Original_Thinking   | 93.2      | 3956         | 56.7      | 12102        | 49.5      | 8292         | -        | -          |
> | Original_NoThinking | 79.6      | 723          | 23.3      | 2921         | 12.1      | 714          | -28.1    | -83.0%     |
> | Fast-Solving Prompt | 88.8      | 2738         | 50.0      | 12910        | **53.0**  | 7992         | -2.5     | -9.2%      |
> | SFT                 | 89.8      | 3912         | 43.3      | 9799         | 48.0      | 10437        | -6.1     | +1.9%      |
> | DPO                 | 94.0      | 3227         | 56.7      | 9514         | 52.0      | 9056         | +1.1     | -10.2%     |
> | O1-Pruner           | **94.8**  | 3895         | **63.3**  | 11401        | 49.5      | 8731         | **+2.7** | -0.7%      |
> | AdaptThink          | 92.6      | **2032**     | 56.7      | **9214**     | 51.5      | 7336         | +0.5     | **-28.0%** |
> | Ours                | 92.4      | 3105         | **63.3**  | 10230        | 49.0      | **6626**     | +1.8     | -19.0%     |
>
> ### Reply to Weakness 7: Relationship Between Problem Difficulty, Entropy, and Overthinking
>
> > A significant concern is that the relationship between problem difficulty, entropy, and model overthinking is not sufficiently clarified. While some correlation experiments are presented, they are not comprehensive enough. It would be valuable to see more detailed analyses or theoretical proofs addressing these interconnections.
>
> In Appendices A.2 and A.3, we demonstrate the relationship between problem difficulty, average entropy, and overthinking through analysis of responses at different difficulty gradients on the MATH dataset and across datasets. Experimental results show that as problem difficulty increases, model output length increases substantially and the proportion of high-entropy tokens in model responses rises significantly. These results all explain why average entropy can reflect problem difficulty. If further experiments are needed, please specify.
>
> ### References
>
> [1] The Entropy Mechanism of Reinforcement Learning for Reasoning Language Models
>
> [2] Beyond the 80/20 Rule: High-Entropy Minority Tokens Drive Effective Reinforcement Learning for LLM Reasoning
>
> [3] REASONING WITH EXPLORATION: AN ENTROPY PER-SPECTIVE ON REINFORCEMENT LEARNING FOR LLMS
>
> [4] Token-Budget-Aware LLM Reasoning
>
> [5] CoT-Valve: Length-Compressible Chain-of-Thought Tuning

---

> ### Author Response · Authors · 2025-11-27
> **Gentle Reminder to Reviewer 1kVB**
>
> Dear Reviewer 1kVB,
>
> We greatly appreciate the time and effort you've taken to review our submission. We hope that our response has addressed the concerns raised in your initial reviews, and we look forward to your feedback.
>
> Since the discussion window has less than a week remaining, we sincerely look forward to your reevaluation of our work. **If our response has successfully addressed your concerns, we would greatly appreciate it if you could consider raising your score** to boost our chance of more exposure to the community. Thank you very much!
>
> Please let us know if you need any further information or clarification.
>
> Best regards and thanks,
>
> Paper 11598 Authors

---

### Official Review · Reviewer_Ddkx · 2025-10-31

**Soundness:** 2
**Presentation:** 2
**Contribution:** 2
**Rating:** 4
**Confidence:** 4

**Summary:**

This paper introduces an entropy-driven training framework to reduce overthinking in large language models. By leveraging token-level entropy as a measure of problem complexity, the model learns to choose between concise NoThinking and full Thinking modes. A two-stage training process, which leverages supervised fine-tuning followed by offline PPO, teaches the model to adaptively allocate reasoning effort. Experiments on MATH500, AIME24, and GPQA show significant reductions in response length with maintained or improved accuracy across multiple model sizes.

**Strengths:**

This paper identifies token-level entropy as an intrinsic, continuous measure of problem complexity. This is more fundamental and fine-grained than previous heuristics, such as response length, providing a reasonable signal for adaptive reasoning.

**Weaknesses:**

1.The evaluation primarily centers on mathematical reasoning benchmarks, which, while offering clear distinctions in difficulty levels that align well with your approach, raises questions about generalizability. The broader applicability of the entropy-complexity relationship and the effectiveness of the proposed training framework to other domains, such as commonsense reasoning, remains to be verified.
2.The experiments utilize models distilled from DeepSeek-R1. While such distilled models inherently possess stronger reasoning capabilities, it remains unclear whether the original non-distilled models would yield consistent results with your current experimental findings. Furthermore, the performance of other model architectures, such as GLM and Mistral, is also not yet verified.
3.The current work lacks discussion on the multi-component base reward. The experiments omit the exploration of the weighting scheme assigned to the three distinct rewards.

**Questions:**

1.There are issues with Figure 1 on page 2. First, the "simple question" is positioned at the top of the figure, which is mistaken in the caption. Second, there is an inconsistency: the "simple question" is listed as "What is 50% of 240?" but is solved as "15% of 240" in the subsequent steps.
2.There is a typo in Figure 2: "deepseek15B" should be corrected to "deepseek1.5B".
Page 4, Line 198: It is recommended to provide the full, official name of the model. The current designation "deepseek-1.5B" is ambiguo

---

> ### Author Response · Authors · 2025-11-18
> **Official Response to Reviewer Ddkx Part (1/2)**
>
> ### Reply to Weakness 1: Evaluation Beyond Mathematical Reasoning
>
> > The evaluation primarily centers on mathematical reasoning benchmarks, which, while offering clear distinctions in difficulty levels that align well with your approach, raises questions about generalizability. The broader applicability of the entropy-complexity relationship and the effectiveness of the proposed training framework to other domains, such as commonsense reasoning, remains to be verified.
>
> Thank you for your valuable suggestion. In other domains such as creative writing and commonsense reasoning, problem difficulty is challenging to measure. Considering the differences between large models and humans in knowledge retrieval and reasoning, it is difficult to find datasets with clear difficulty gradients similar to the MATH dataset for verification, so we only explored the entropy-complexity correlation in mathematics. However, even though our training data only includes mathematical content, we also tested on broader evaluation domains such as GPQA and MMLU during model testing. Experimental results demonstrate that the processed model exhibits good performance on these datasets as well, sufficiently demonstrating the good generalizability of our method.
>
> ### Reply to Weakness 2: Experiments on Original Models
>
> > The experiments utilize models distilled from DeepSeek-R1. While such distilled models inherently possess stronger reasoning capabilities, it remains unclear whether the original non-distilled models would yield consistent results with your current experimental findings. Furthermore, the performance of other model architectures, such as GLM and Mistral, is also not yet verified.
>
> Our work focuses on addressing the over-thinking problem in long reasoning chains (with `<think>` thinking content). Original models that do not adopt long CoT mode are not within the scope of our research. As of our experimental timeframe, only DeepSeek small models based on Qwen and Llama architectures had open-source versions of reasoning models. We believe that conducting experiments on these two architectures sufficiently demonstrates the good generalizability of our method across model architectures.

---

> ### Author Response · Authors · 2025-11-18
> **Official Response to Reviewer Ddkx Part (2/2)**
>
> ### Reply to Weakness 3: Exploration of Multi-Component Base Rewards
>
> In ablation experiments, we ablated the effectiveness of our proposed difficulty-aware reward component, and experimental results demonstrated the effectiveness of our proposed component. The influence of other components of the reward function has been discussed in baseline work.
>
> Due to unavailability of previous training resources, we switched to different training GPU resources, which led to partial inconsistencies between optimal hyperparameters and results compared to the original manuscript. However, the overall performance differences are minimal. We have listed the differences (from left to right, Acc and Length correspond to three datasets: MATH500, AIME2024, GPQA, aligned with columns in Table 1 of the paper. The hyperparameters correspond to correctness reward coefficient, length penalty coefficient, and difficulty-adaptive coefficient. For readability, we use one decimal place for distinction).
>
> | Hyperparameters | MATH-Acc↑ | MATH-Length↓ | AIME-Acc↑ | AIME-Length↓ | GPQA-Acc↑ | GPQA-Length↓ |
> | --------------- | --------- | ------------ | --------- | ------------ | --------- | ------------ |
> | origin          | 92.4      | 3105         | 63.3      | 10230        | 49.0      | 6626         |
> | 1-5-3           | 89.2      | 3315         | 63.3      | 10383        | 50.0      | 6824         |
> | 3-3-3           | 92.2      | 3205         | 56.7      | 11793        | 46.0      | 6773         |
> | 3-5-1           | 91.0      | 3331         | 53.3      | 13643        | 49.0      | 6655         |
> | 3-5-3           | 91.6      | 3254         | 56.7      | 12002        | 52.5      | 6467         |
> | 3-5-5           | 90.4      | 3108         | 56.7      | 12570        | 54.0      | 6679         |
> | 3-7-3           | 90.6      | 3128         | 60.0      | 10896        | 48.5      | 6536         |
> | 5-3-3           | 90.0      | 3240         | 53.3      | 10863        | 47.5      | 6794         |
> | 5-3-5           | 92.2      | 3380         | 53.3      | 9458         | 49.0      | 6676         |
> | 5-5-3 *         | 91.4      | 3148         | 63.3      | 9815         | 49.0      | 6830         |
> | 5-5-5           | 89.6      | 3268         | 46.7      | 12279        | 48.5      | 6461         |
> | 7-5-3           | 90.2      | 3252         | 63.3      | 12763        | 53.0      | 6612         |
>
> Based on the experimental results, the three hyperparameters exhibit different impact patterns on model performance:
>
> The influence of the correctness reward coefficient ($w_1$) presents nonlinear characteristics. For in-domain datasets, medium-low values ($w_1$=0.3-0.5) can achieve optimal accuracy (around 92%), but for out-of-domain high-difficulty tasks, higher correctness rewards ($w_1$=0.7) show advantages, particularly achieving 63.3% accuracy on the AIME dataset. This indicates that the role of correctness reward is task-dependent—simple tasks do not require overly strong correctness incentives, while complex reasoning tasks benefit from stronger correctness guidance.
>
> The length penalty coefficient ($w_2$) demonstrates clear and stable control over output efficiency. When $w_2$=0.5, the model produces more concise outputs across all datasets. For example, on the AIME dataset, configurations with $w_2$=0.5 average around 10,000 words, significantly shorter than configurations with $w_2$=0.3. Interestingly, moderate length penalty ($w_2$=0.5) does not harm accuracy; rather, it improves out-of-domain generalization in some configurations, possibly because enforcing concise expression compels the model to learn more essential reasoning patterns.
>
> The difficulty-adaptive coefficient ($w_3$) exhibits interactions with task difficulty. Medium values ($w_3$=0.3) perform best on the high-difficulty AIME dataset (multiple configurations achieve 63.3%), while higher values ($w_3$=0.5) show slight advantages on the medium-difficulty GPQA. This "counter-intuitive" phenomenon suggests that excessive difficulty adaptation may lead the model to over-adjust strategies on truly difficult tasks, while moderate adaptation ($w_3$=0.3) provides more robust performance.
>
> There exists synergy among the three hyperparameters: the balanced configuration of 5-5-3 exhibits the best comprehensive performance, indicating that in reinforcement learning fine-tuning, moderate balance across dimensions is more important than extreme optimization of any single dimension. This finding has important practical implications—pursuing overall balanced reward design cultivates model robustness and generalization capability better than emphasizing any single aspect.
>
> ### Reply to Question 1 and 2: Figure Issues and Spelling Errors
>
> Thank you for pointing out these issues. In the final revised version, we will correct figure formatting and spelling errors.

---

> ### Author Response · Authors · 2025-11-27
> **Gentle Reminder to Reviewer Ddkx**
>
> Dear Reviewer Ddkx,
>
> We greatly appreciate the time and effort you've taken to review our submission. We hope that our response has addressed the concerns raised in your initial reviews, and we look forward to your feedback.
>
> Since the discussion window has less than a week remaining, we sincerely look forward to your reevaluation of our work. **If our response has successfully addressed your concerns, we would greatly appreciate it if you could consider raising your score** to boost our chance of more exposure to the community. Thank you very much!
>
> Please let us know if you need any further information or clarification.
>
> Best regards and thanks,
>
> Paper 11598 Authors

---

### Official Review · Reviewer_Hkfd · 2025-11-01

**Soundness:** 3
**Presentation:** 3
**Contribution:** 3
**Rating:** 6
**Confidence:** 4

**Summary:**

**Summary:**
The paper studies how LLMs tend to overthink by producing unnecessarily long reasoning chains even for simple problems. It identifies response entropy as a reliable indicator of model confidence and problem complexity, showing that entropy correlates with task difficulty. Building on this insight, the authors propose a two-stage framework: supervised fine-tuning for direct answering on easy cases, followed by offline PPO with a confidence reward to encourage adaptive reasoning. Experiments on multiple reasoning benchmarks demonstrate reduced reasoning length with competitive or improved accuracy.

**Contributions:**

1. Systematically validate that response entropy serves as an effective intrinsic measure of problem complexity in LLM reasoning.
2. Introduce a confidence reward based on response entropy during the PPO phase, with ablation results showing that this design improves performance and reduces generation length.

**Strengths:**

**Originality:** The paper offers a novel perspective by validating *response entropy* as an intrinsic measure of problem complexity and incorporating it into offline PPO via a confidence reward.

**Quality:** The analysis and experiments are comprehensive and convincing, with ablation results clearly supporting the proposed design.

**Clarity:** The paper is clearly written, well organized, and provides sufficient implementation details for reproducibility.

**Significance:** The findings provide practical insights for improving reasoning efficiency and adaptability in LLMs.

**Weaknesses:**

1. **Comparison with alternative adaptive reasoning methods:** The paper could include stronger baselines, such as other adaptive reasoning strategies (e.g., AdaptThink [1], AdaCoT [2]).
2. **Discussion of internal uncertainty and problem difficulty:** The paper lacks sufficient discussion of prior studies on how internal uncertainty relates to problem difficulty (e.g., [3]), which would better situate the contribution within existing literature.

**References:**

[1] AdaptThink: Reasoning Models Can Learn When to Think, [https://arxiv.org/pdf/2505.13417](https://arxiv.org/pdf/2505.13417)

[2] AdaCoT: Pareto-Optimal Adaptive Chain-of-Thought Triggering via Reinforcement Learning, [https://arxiv.org/pdf/2505.11896](https://arxiv.org/pdf/2505.11896)

[3] Think or Not? Exploring Thinking Efficiency in Large Reasoning Models via an Information-Theoretic Lens, [https://arxiv.org/pdf/2505.18237](https://arxiv.org/pdf/2505.18237)

**Questions:**

1. **Reward weighting:** In formula (4), how were the weights for the different reward components determined? Specifically, why is the accuracy reward set to 0.3 rather than being the largest among the three? Were alternative weight configurations explored, and if so, what were the outcomes?

2. **Design of the confidence reward:** The confidence reward promotes higher certainty in non-thinking mode and lower certainty in thinking mode. Could the authors clarify the rationale for this design? In formula (7), the reward appears to be applied only when the generated answer is correct. If the answer is correct, why not always encourage higher confidence? Conversely, why is the reward set to zero when the answer is incorrect?

3. **Correlation in Figure 2:** How is the “Correlation with Difficulty” on the y-axis of Figure 2 computed?

---

> ### Author Response · Authors · 2025-11-18
> **Official Response to Reviewer Hkfd Part (1/3)**
>
> ### Reply to Weakness 1: Comparison with Other Adaptive Reasoning Methods
>
> > The paper could include stronger baselines, such as other adaptive reasoning strategies (e.g., AdaptThink [1], AdaCoT [2]).
>
> Thank you for your valuable suggestion. Due to limited computational resources, we were unable to directly reproduce AdaptThink (requiring 32×A100) and other methods requiring substantial computational resources. O1-Pruner was the only comparison method we found that uses the same offline reinforcement learning training process. However, since AdaptThink provides open-source models, we present comparison experimental results based on their open-source models AdaptThink-1.5B-delta0.05 and AdaptThink-7B-delta0.05 as follows (from left to right, Acc and Length correspond to three datasets: MATH500, AIME2024, GPQA, and Average, aligned with columns in Table 1 of the paper). We will include these experimental results in the revised version. We would also like to note that compared to AdaptThink's online reinforcement learning approach, the relatively lower training cost is one of the advantages of our method.
>
> **DeepSeek-R1-Distill-Qwen-1.5B**
>
> | Method                       | MATH-Acc↑ | MATH-Length↓ | AIME-Acc↑ | AIME-Length↓ | GPQA-Acc↑ | GPQA-Length↓ | ΔAcc↑    | ΔLength↓   |
> | ---------------------------- | --------- | ------------ | --------- | ------------ | --------- | ------------ | -------- | ---------- |
> | Original$_\text{Thinking}$   | 85.2      | 5461         | 23.3      | 17231        | 39.4      | 9405         | -        | -          |
> | Original$_\text{NoThinking}$ | 70.4      | 944          | 10.0      | 2635         | 13.6      | 782          | -18.0    | -86.4%     |
> | Fast-Solving Prompt          | 78.2      | 3344         | 26.7      | 14424        | 36.9      | 9312         | -2.0     | -18.7%     |
> | SFT                          | 80.6      | 4965         | 23.3      | 17575        | 32.8      | 10322        | -3.7     | +0.9%      |
> | DPO                          | **84.8**  | 4816         | **33.3**  | 14219        | 34.8      | 9556         | +1.7     | -9.2%      |
> | O1-Pruner                    | 82.8      | 4635         | 30.0      | 16841        | 37.9      | 8869         | +0.9     | -7.7%      |
> | AdaptThink                   | 84.4      | **1985**     | 30.0      | 9192         | 35.3      | 7854         | +0.6     | -42.3%     |
> | Ours                         | 84.2      | 3257         | **33.3**  | **7250**     | **39.4**  | **5260**     | **+3.0** | **-47.5%** |
>
> **DeepSeek-R1-Distill-Qwen-7B**
>
> | Method                       | MATH-Acc↑ | MATH-Length↓ | AIME-Acc↑ | AIME-Length↓ | GPQA-Acc↑ | GPQA-Length↓ | ΔAcc↑    | ΔLength↓   |
> | ---------------------------- | --------- | ------------ | --------- | ------------ | --------- | ------------ | -------- | ---------- |
> | Original$_\text{Thinking}$   | 93.2      | 3956         | 56.7      | 12102        | 49.5      | 8292         | -        | -          |
> | Original$_\text{NoThinking}$ | 79.6      | 723          | 23.3      | 2921         | 12.1      | 714          | -28.1    | -83.0%     |
> | Fast-Solving Prompt          | 88.8      | 2738         | 50.0      | 12910        | **53.0**  | 7992         | -2.5     | -9.2%      |
> | SFT                          | 89.8      | 3912         | 43.3      | 9799         | 48.0      | 10437        | -6.1     | +1.9%      |
> | DPO                          | 94.0      | 3227         | 56.7      | 9514         | 52.0      | 9056         | +1.1     | -10.2%     |
> | O1-Pruner                    | **94.8**  | 3895         | **63.3**  | 11401        | 49.5      | 8731         | **+2.7** | -0.7%      |
> | AdaptThink                   | 92.6      | **2032**     | 56.7      | **9214**     | 51.5      | 7336         | +0.5     | **-28.0%** |
> | Ours                         | 92.4      | 3105         | **63.3**  | 10230        | 49.0      | **6626**     | +1.8     | -19.0%     |
>
> ### Reply to Weakness 2: Discussion of Internal Uncertainty
>
> > The paper lacks sufficient discussion of prior studies on how internal uncertainty relates to problem difficulty (e.g., [3]), which would better situate the contribution within existing literature.
>
> Thank you for your valuable suggestion. The paper you mentioned, "Think or Not? Exploring Thinking Efficiency in Large Reasoning Models via an Information-Theoretic" approach, investigates information gain during model reasoning and proposes a method to dynamically control reasoning termination based on entropy of the answer space. Our method focuses on the post-training stage. Additionally, our analysis of token-level entropy in model performance encompasses the entire reasoning process and all token distribution spaces, revealing that models exhibit a higher proportion of high-entropy tokens in their outputs when facing problems with greater uncertainty. We will include the above discussion in the revised version.

---

> ### Author Response · Authors · 2025-11-18
> **Official Response to Reviewer Hkfd Part (2/3)**
>
> ### Reply to Question 1: Determination of Reward Weights
>
> > In formula (4), how were the weights for the different reward components determined? Specifically, why is the accuracy reward set to 0.3 rather than being the largest among the three? Were alternative weight configurations explored, and if so, what were the outcomes?
>
> Due to unavailability of previous training resources, we switched to different training GPU resources, which led to partial inconsistencies between optimal hyperparameters and results compared to the original manuscript. However, the overall performance differences are minimal. We have listed the differences (from left to right, Acc and Length correspond to three datasets: MATH500, AIME2024, GPQA, aligned with columns in Table 1 of the paper. The hyperparameters correspond to correctness reward coefficient, length penalty coefficient, and difficulty-adaptive coefficient. For readability, we use one decimal place for distinction).
>
> | Hyperparameters | MATH-Acc↑ | MATH-Length↓ | AIME-Acc↑ | AIME-Length↓ | GPQA-Acc↑ | GPQA-Length↓ |
> | --------------- | --------- | ------------ | --------- | ------------ | --------- | ------------ |
> | origin          | 92.4      | 3105         | 63.3      | 10230        | 49.0      | 6626         |
> | 1-5-3           | 89.2      | 3315         | 63.3      | 10383        | 50.0      | 6824         |
> | 3-3-3           | 92.2      | 3205         | 56.7      | 11793        | 46.0      | 6773         |
> | 3-5-1           | 91.0      | 3331         | 53.3      | 13643        | 49.0      | 6655         |
> | 3-5-3           | 91.6      | 3254         | 56.7      | 12002        | 52.5      | 6467         |
> | 3-5-5           | 90.4      | 3108         | 56.7      | 12570        | 54.0      | 6679         |
> | 3-7-3           | 90.6      | 3128         | 60.0      | 10896        | 48.5      | 6536         |
> | 5-3-3           | 90.0      | 3240         | 53.3      | 10863        | 47.5      | 6794         |
> | 5-3-5           | 92.2      | 3380         | 53.3      | 9458         | 49.0      | 6676         |
> | 5-5-3 *         | 91.4      | 3148         | 63.3      | 9815         | 49.0      | 6830         |
> | 5-5-5           | 89.6      | 3268         | 46.7      | 12279        | 48.5      | 6461         |
> | 7-5-3           | 90.2      | 3252         | 63.3      | 12763        | 53.0      | 6612         |
>
> Based on the experimental results, the three hyperparameters exhibit different impact patterns on model performance:
>
> The influence of the correctness reward coefficient ($w_1$) presents nonlinear characteristics. For in-domain datasets, medium-low values ($w_1$=0.3-0.5) can achieve optimal accuracy (around 92%), but for out-of-domain high-difficulty tasks, higher correctness rewards ($w_1$=0.7) show advantages, particularly achieving 63.3% accuracy on the AIME dataset. This indicates that the role of correctness reward is task-dependent—simple tasks do not require overly strong correctness incentives, while complex reasoning tasks benefit from stronger correctness guidance.
>
> The length penalty coefficient ($w_2$) demonstrates clear and stable control over output efficiency. When $w_2$=0.5, the model produces more concise outputs across all datasets. For example, on the AIME dataset, configurations with $w_2$=0.5 average around 10,000 words, significantly shorter than configurations with $w_2$=0.3. Interestingly, moderate length penalty ($w_2$=0.5) does not harm accuracy; rather, it improves out-of-domain generalization in some configurations, possibly because enforcing concise expression compels the model to learn more essential reasoning patterns.
>
> The difficulty-adaptive coefficient ($w_3$) exhibits interactions with task difficulty. Medium values ($w_3$=0.3) perform best on the high-difficulty AIME dataset (multiple configurations achieve 63.3%), while higher values ($w_3$=0.5) show slight advantages on the medium-difficulty GPQA. This "counter-intuitive" phenomenon suggests that excessive difficulty adaptation may lead the model to over-adjust strategies on truly difficult tasks, while moderate adaptation ($w_3$=0.3) provides more robust performance.
>
> There exists synergy among the three hyperparameters: the balanced configuration of 5-5-3 exhibits the best comprehensive performance, indicating that in reinforcement learning fine-tuning, moderate balance across dimensions is more important than extreme optimization of any single dimension. This finding has important practical implications—pursuing overall balanced reward design cultivates model robustness and generalization capability better than emphasizing any single aspect.

---

> ### Author Response · Authors · 2025-11-18
> **Official Response to Reviewer Hkfd Part (3/3)**
>
> ### Reply to Question 2: Design of Confidence Reward
>
> > The confidence reward promotes higher certainty in non-thinking mode and lower certainty in thinking mode. Could the authors clarify the rationale for this design? In formula (7), the reward appears to be applied only when the generated answer is correct. If the answer is correct, why not always encourage higher confidence? Conversely, why is the reward set to zero when the answer is incorrect?
>
> Thank you for your valuable question. Our reinforcement learning step aims to encourage the model to use the non-thinking mode as much as possible while maintaining accuracy. Specifically, in direct answer mode, rewards increase with certainty, whereas in thinking mode, rewards increase with uncertainty. This achieves the goal of encouraging confident answers in non-thinking mode and uncertain answers in thinking mode, making responses more aligned with the characteristics of these two modes. When answers are incorrect, the model may need longer reasoning to attempt obtaining correct answers. However, some research also indicates that the lack of thinking process is not the reason for incorrect model answers \[1][2], and shorter thinking processes may help models provide accurate answers. Given that current research in this area lacks definitive conclusions, to simplify our method, we set rewards to 0 when model answers are incorrect to avoid introducing uncertain noise. The reward setting when answers are incorrect is an important research direction, which we will explore further in future work.
>
> ### Reply to Question 3: Correlation in Figure 2
>
> > Correlation in Figure 2: How is the “Correlation with Difficulty” on the y-axis of Figure 2 computed?
>
> We used the Pearson correlation coefficient as the method to calculate the correlation between the custom difficulty correlation coefficient D and the filtered average entropy. We will provide detailed explanation in the revised version.
>
> ### References
> [1] AdaptThink: Reasoning Models Can Learn When to Think, https://arxiv.org/pdf/2505.13417
>
> [2] O1-Pruner: Length-Harmonizing Fine-Tuning for O1-Like Reasoning, https://arxiv.org/abs/2501.12570

---

> ### Comment · Reviewer_Hkfd · 2025-11-19
> **Response to authors**
>
> Thank you for the additional experimental results and clarifications. Your response has fully addressed my concerns, and I will maintain my positive evaluation.

---

### Official Review · Reviewer_qXEm · 2025-11-01

**Soundness:** 3
**Presentation:** 2
**Contribution:** 2
**Rating:** 4
**Confidence:** 4

**Summary:**

This work tackles the overthinking problem in LLMs, where they produce unnecessarily long reasoning chains for simple tasks. The authors identify token-level response entropy as an intrinsic indicator of problem complexity and use it to guide adaptive reasoning. They propose a two-stage training framework: (1) Supervised fine-tuning on concise “NoThinking” exemplars to enable short, confident answers, and (2) offline PPO with an entropy-aware reward to let the model choose between concise and full reasoning based on task difficulty. Experiments on MATH500, AIME24, and GPQA show that this method reduces response length while maintaining accuracy, confirming entropy’s value as both a diagnostic signal and a training objective for efficient reasoning in LLMs.

**Strengths:**

1. Letting LLMs not over-think on simple questions is an important question with practical applications
2. The finding of the entropy of response correlates well with problem difficulty is interesting.
3. The results of reducing response lengths while keeping accuracy are good.

**Weaknesses:**

1. The definition of problem difficulty needs further clarification and justification.
2. For base model selection, why not consider Qwen-2.5/3 reasoning models or other alternatives?
3. Discuss how this method compares with the GPT-5 router and what advantages it offers.

**Questions:**

Please see above weaknesses.

---

> ### Author Response · Authors · 2025-11-14
> **Official Response to Reviewer qXEm**
>
> ### Reply to Weakness 1: Definition of Problem Difficulty
> > The definition of problem difficulty needs further clarification and justification.
>
> Thank you very much for your valuable question. Regarding the definition of problem difficulty: we reference the manually annotated difficulty coefficients in the MATH dataset and the source of the problems. Based on large model performance, we interpret problem difficulty as the computational complexity required for model solutions and the uncertainty in model responses. This manifests in metrics like answer length and accuracy, detailed in Appendix A.1 of our paper, which explores model performance under human-like cognitive challenges. We will further elaborate and clarify this definition in the revised version.
>
> ###  Reply to Weakness 2: Model Selection
> > For base model selection, why not consider Qwen-2.5/3 reasoning models or other alternatives?
>
> Thank you for your valuable question. Regarding model series selection, the DeepSeek series comprises models widely adopted in both academia and industry. The Qwen2.5 models (including Math and Coder variants) you mentioned predate DeepSeek and lack corresponding inference models, while Qwen3-4B models emerged after August 2025. Considering the timeline for paper experimentation and writing, we selected more mature and widely adopted models. Additionally, DeepSeek offers two small models distilled from Qwen and Llama architectures. Conducting experiments on both architectures effectively demonstrates the versatility and portability of our methodology.
>
> ###  Reply to Weakness 3: Comparison with GPT-5 Router
> > Discuss how this method compares with the GPT-5 router and what advantages it offers.
>
> Thank you for your valuable question. To our knowledge, GPT-5's router selects different LLMs based on user query style and tool invocation requirements—a different objective from our single-model processing approach. Moreover, GPT-5 is a closed-source, large-scale model whose technical details are difficult to verify. We provide a concrete, open-source processing method tailored for smaller-scale models, which offers greater value to the research community.

---

> ### Author Response · Authors · 2025-11-27
> **Gentle Reminder to Reviewer qXEm**
>
> Dear Reviewer qXEm,
>
> We greatly appreciate the time and effort you've taken to review our submission. We hope that our response has addressed the concerns raised in your initial reviews, and we look forward to your feedback.
>
> Since the discussion window has less than a week remaining, we sincerely look forward to your reevaluation of our work. **If our response has successfully addressed your concerns, we would greatly appreciate it if you could consider raising your score** to boost our chance of more exposure to the community. Thank you very much!
>
> Please let us know if you need any further information or clarification.
>
> Best regards and thanks,
>
> Paper 11598 Authors

---

### Meta-Review · Area_Chair_2LTj · 2026-01-07

**Summary:**

This paper investigates the overthinking behavior of LLMs and proposes an entropy-driven training framework that adaptively allocates reasoning effort. While the problem is relevant and the empirical results demonstrate consistent reductions in response length without harming accuracy, the contribution is primarily empirical and incremental in nature. The rebuttal successfully addressed several technical concerns by adding missing baselines, reward ablations, and clarifications around optimization terminology. However, the work remains narrowly evaluated on mathematical reasoning benchmarks and relies heavily on distilled DeepSeek-R1 models, limiting confidence in its generality. The relationship between entropy, problem difficulty, and overthinking is validated empirically but lacks a strong theoretical grounding. Overall, the paper is technically sound but falls short of the depth, breadth, and clarity expected for a top-tier venue.

**Reviewer Concerns:**

Addressed concerns -

* Lack of strong adaptive reasoning baselines
* Insufficient reward ablation analysis
* Related work coverage

Outstanding concerns -

* Evaluation is largely confined to math reasoning tasks; applicability to broader reasoning domains is not demonstrated.
* Dependence on distilled DeepSeek-R1 models raises concerns about transferability to standard, non-distilled LLMs.
* The entropy, difficulty, overthinking connection remains correlational, with limited theoretical justification.

**Reviewer Scores:**

Reviewer 1kVB was largely unresponsive throughout the period. I firmly believe that his/her score would have improved by 2 points given the rebuttal responses.

---

### Decision · Program_Chairs · 2026-01-26

Reject